# Fermi-gas correlators of ADHM theory and triality symmetry

**Yasuyuki Hatsuda[1*] and Tadashi Okazaki[2†]**

**1** Department of Physics, Rikkyo University, Toshima, Tokyo 171-8501, Japan
**2** Department of Mathematical Sciences, Durham University,
Upper Mountjoy Campus, Stockton Road, Durham DH1 3LE, UK

\* yhatsuda@rikkyo.ac.jp   † tadashi.okazaki@durham.ac.uk

## Abstract

We analytically study the Fermi-gas formulation of sphere correlation functions of the Coulomb branch operators for 3d $\mathcal{N} = 4$ ADHM theory with a gauge group $U(N)$, an adjoint hypermultiplet and $l$ hypermultiplets which can describe a stack of $N$ M2-branes at $A_{l-1}$ singularities. We find that the leading coefficients of the perturbative grand canonical correlation functions are invariant under a hidden triality symmetry conjectured from the twisted M-theory. The triality symmetry also helps us to fix the next-to-leading corrections analytically.


# 1  Introduction

The technique of supersymmetric localization allows for exact computations of observables in quantum field theories with a certain minimum amount of supersymmetry as pioneered by Pestun [1] (see [2] for reviews).

It was shown in [3] that the partition function on a three-sphere $S^3$ of a three-dimensional gauge theory with $\mathcal{N} \geq 3$ supersymmetry reduces to a matrix model via the localization.

For three-dimensional theories with $\mathcal{N} = 4$ supersymmetry the sphere partition functions can be decorated by either of two types of half-BPS local operators; the Coulomb (resp. Higgs) branch operators whose expectation values define the Coulomb (resp. Higgs) branch. As shown in [4–6], the localization also allows for the evaluation of these protected correlators as generalized matrix integrals in such a way that a collection of the Coulomb or Higgs branch operators localize along a specific great circle $S^1$ in the $S^3$.

The $\Omega$ deformation is very useful in the study of the supersymmetric gauge theories (See e.g. [7–12]). The protected correlation functions of the Higgs branch operators or the Coulomb branch operators in 3d $\mathcal{N} = 4$ supersymmetric field theories are encoded by one-dimensional topological quantum mechanical models [13, 14]. The associated topological quantum mechanical models arise from certain $\Omega$ deformations of the parent three-dimensional $\mathcal{N} = 4$ supersymmetric field theories [15–17]. The topological quantum mechanics can be viewed as a non-trivial space of solution to OPE Ward identities equipped with the quantized Coulomb (resp. Higgs) branch algebra [15, 18] which results from the quantization of the chiral ring of the Coulomb (resp. Higgs) branch operators.

It is shown in [19] that these sphere correlation functions of the Coulomb or Higgs branch operators as well as the sphere partition functions can be algebraically presented from the quantized Coulomb and Higgs branch algebras in terms of the twisted traces over the Verma modules without relying on the UV data.

The 3d $\mathcal{N} = 4$ superconformal field theory (SCFT) appearing at low energy on a stack of $N$ M2-branes on $\mathbb{R} \times \mathbb{C}$ at an $A_{l-1}$ singularity probing the space $\mathbb{C}^2 \times (\mathbb{C}^2/\mathbb{Z}_l)$ has a UV description as a 3d $\mathcal{N} = 4$ ADHM gauge theory with a gauge group $U(N)$, an adjoint hypermultiplet and $l$ fundamental hypermultiplets [20, 21]. In the near horizon limit of the M2-branes, one obtains the holographic dual $AdS_4 \times (S^7/\mathbb{Z}_l)$ background of M-theory, which provides us with an attractive example of the AdS/CFT correspondence [22–24]. For $l = 1$, the ADHM theory is self-mirror [25, 26] and equivalent to the ABJM theory with $k = 1$ in the IR [27].

The large $N$ limit of the partition function in the ADHM theory was studied well in [28, 29]. In spite of the explicit expressions of the partition function, it is still tricky to evaluate it analytically in the large $N$ regime. One of the analytic approach for the partition function is the Fermi-gas formalism [30] where the partition function is rewritten as the partition function of an ideal Fermi gas of $N$ non-interacting particles. In the large $N$ expansion of the free energy $F = -\log Z_{S^3}$ for the SCFT of the $N$ M2-branes where $Z_{S^3}$ is the sphere partition function, the leading coefficient can be evaluated from the two-derivative supergravity [31, 32] and the next-to-leading coefficient is expected to be reproduced from higher derivative corrections in

the supergravity. [1]

Recently it has been proposed in [34] that the topological quantum mechanics that encodes the protected correlation functions of the world-volume theory of M2-branes are holographically dual to a certain protected sector of M-theory, that is the $\Omega$-deformed or topologically twisted M-theory as an example of "twisted holography" [35].

Topologically twisted M-theory on an $\Omega$-deformed background

$$\mathbb{R} \times \mathbb{C}^2/\mathbb{Z}_l \times \mathbb{C}_{\epsilon_1} \times \mathbb{C}_{\epsilon_2} \times \mathbb{C}_{\epsilon_3}, \tag{1}$$

obeying the Calabi-Yau condition $\epsilon_1 + \epsilon_2 + \epsilon_3 = 0$ can lead to a 5d theory on $\mathbb{R} \times \mathbb{C}^2/\mathbb{Z}_l$ [36,37]. It is called the "twisted M-theory". The twisted M-theory is locally trivial and it is topological in $\mathbb{R}$ and holomorphic in the remaining four directions. It depends on the ratio $\epsilon_2/\epsilon_1$ and it has a perturbative description as a non-commutative Chern-Simons theory at least for $l = 1$ and in some range of parameters. One interesting feature of this theory is a triality symmetry that permutes the $\Omega$-deformation parameters [38]

$$\epsilon_1 \to \epsilon_2, \qquad\qquad \epsilon_2 \to \epsilon_3, \qquad\qquad \epsilon_3 \to \epsilon_1. \tag{2}$$

The twisted M-theory can contain M2-branes and M5-branes as line operators and surface operators in the 5d theory.[2] In the $\Omega$-deformed background (1), when a stack of $N$ M2-branes are placed on $\mathbb{R} \times \mathbb{C}_{\epsilon_1}$, the ADHM theory would acquire the mass parameter $m$ for the adjoint hypermultiplet given by

$$m = i\left(\frac{1}{2} + \frac{\epsilon_2}{\epsilon_1}\right) \tag{3}$$

and it can be effectively described at low energy as topological quantum mechanics on $\mathbb{R}$ equipped with certain spherical part of the cyclotomic rational Cherednik algebras [42] with $\epsilon_1$ being the quantization parameter as the quantized Coulomb branch algebra. The perturbative part of the protected correlation functions in the ADHM theory is expected to be holographically dual to a perturbative twisted M-theory background. It is shown from numerical and algebraic calculations in [43] that the perturbative part of protected correlation functions of the ADHM theory with $l = 1$ on a three-sphere enjoys the triality symmetry in the large $N$ limit.

In this paper we analytically evaluate the sphere correlators of the Coulomb branch operators for the ADHM theory in the Fermi-gas formulation. As argued in [44], the large $N$ behavior of the correlation functions can be evaluated from averages of many-body operators in the Fermi-gas. The triality symmetry (2) that is associated to the adjoint mass (3) is a key in our analysis. This symmetry should be manifest in the perturbative part of the correlation functions in the large $N$ limit [43]. In fact we show that the leading terms of the perturbative grand canonical correlation functions are generally invariant under the triality symmetry (2). Moreover, the triality symmetry constrains the forms of the perturbative part. In the Fermi-gas formulation, it is technically hard to compute the subleading terms of the perturbative part analytically. We are able to obtain it partially. The triality symmetry enables us to reconstruct the remaining missing pieces. In this way, we obtain consistent triality invariant subleading terms for higher-point functions.

The organization of this paper is as follows. In the next section, we review the Fermi-gas formulation by following the original argument in [30]. The technique there is extended to the correlators of the Coulomb branch operators, as shown in Section 3. We will derive the large $N$ behavior for the multi-point correlators. Finally, we will give some remarks on related topics in Section 4.

---

[1]See [33] for a recent approach to the higher derivative corrections from the conformal supergravity.

[2]See [38–41] for recent studies of the operator algebras associated with the intersections of M2 and M5 branes in the twisted M-theory.

## 2 Fermi-gas formulation

We start with a review of the Fermi-gas formulation [30] to analyze partition functions in 3d supersymmetric field theories on $S^3$. In the Fermi-gas formulation, it is more convenient to go to the grand canonical ensemble rather than the canonical one. We show how to derive the grand potential in the large chemical potential limit.

Supersymmetric localization reduces path integrals to matrix models [3]. The partition function of the ADHM theory on $S^3$ takes the form

$$Z_{\text{ADHM}} = \frac{1}{N!} \int \prod_{i=1}^{N} d\sigma_i e^{2\pi i \zeta \sigma_i} \frac{\prod_{i<j} 4 \sinh^2 \pi (\sigma_i - \sigma_j)}{\prod_{i,j=1}^{N} 2 \cosh \pi (\sigma_i - \sigma_j - m) \left( \prod_{i=1}^{N} 2 \cosh \pi \sigma_i \right)^l}, \quad (4)$$

where $m$ is the mass of the adjoint hypermultiplet, and $\zeta$ is the Fayet-Iliopoulos (FI) parameter. For $l = 1$, this is equal to the matrix model of the ABJM theory with $k = 1$ [27].

Making use of the Cauchy identity, one can identify the matrix integral (4) with the canonical partition function of a non-interacting, one-dimensional Fermi-gas with $N$ particles [30]:

$$Z_{\text{ADHM}}(N) = \frac{1}{N!} \sum_{\nu \in S_N} (-1)^{\epsilon(\nu)} \int \prod_{i=1}^{N} d\sigma_i \prod_{i=1}^{N} \rho(\sigma_i, \sigma_{\nu(i)}), \quad (5)$$

where

$$\rho(\sigma_1, \sigma_2) = \frac{e^{\pi i \zeta (\sigma_1 + \sigma_2)}}{(2 \cosh \pi \sigma_1)^{\frac{l}{2}} (2 \cosh \pi (\sigma_1 - \sigma_2 - m))(2 \cosh \pi \sigma_2)^{\frac{l}{2}}}, \quad (6)$$

is the one-particle density matrix in the position representation. This leads to a systematic analysis of the large $N$ limit of the partition function on $S^3$ as a thermodynamic limit of an ideal Fermi-gas. The analysis of the Fermi-gas of the partition function below has appeared in [45]. We generalize it to the computation of some correlation functions, and find new features on a manifestation of the hidden triality symmetry, expected in [38, 43].

The thermodynamic limit of an ideal Fermi-gas can be obtained by considering the one-particle problem in the semi-classical approximation and the $1/N$ corrections to the thermodynamic limit can be obtained by evaluating the quantum corrections to the semi-classical limit. In the following discussion, we consider the grand canonical ensemble, in which the grand potential is introduced by

$$e^{J(\mu)} = \sum_{N=0}^{\infty} e^{\frac{2\pi\mu}{\epsilon_1} N} Z_{\text{ADHM}}(N). \quad (7)$$

Our goal in this section is to derive the large $\mu$ limit of $J(\mu)$ by using the Fermi-gas formulation.

Let $\hat{\sigma}$ and $\hat{p}$ be canonically conjugate operators obeying

$$[\hat{\sigma}, \hat{p}] = i\hbar, \quad (8)$$

where $\hbar = \frac{1}{2\pi}$. Then we can write the density matrix operator as

$$\hat{\rho} = e^{-\frac{1}{2}U(\hat{\sigma})} e^{-T(\hat{p})} e^{-\frac{1}{2}U(\hat{\sigma})}, \quad (9)$$

where

$$U(\sigma) = l \log (2 \cosh \pi \sigma) - 2\pi i \zeta \sigma, \quad (10)$$
$$T(p) = \log (2 \cosh \pi p) - 2\pi i m p, \quad (11)$$

so that the kernel (6) can be realized as the matrix element of the operator (9) in the position space.

From the density matrix operator (9), we can define a one-body Hamiltonian $\hat{H}$ of a system by

$$e^{-\hat{H}} := \hat{\rho}\,. \tag{12}$$

In the classical limit, this Hamiltonian reduces to

$$H_{\mathrm{cl}}(\sigma, p) = U(\sigma) + T(p)\,, \tag{13}$$

where $U(\sigma)$ is a potential term and $T(p)$ is a kinetic term. In the following we analyze the Fermi-gas system in the case where the FI parameter $\zeta$ and the adjoint mass parameter $m$ are pure imaginary so that the Hamiltonian is real. However, we expect that the expression with arbitrary $\zeta$ and $m$ can be reached from our result by analytic continuation.

Given a density matrix (6), one can quantize the Fermi-gas system by following the phase space formulation that is distinguished from the canonical quantization and the path integral formulation. The phase space quantization is based on the Wigner-Weyl transforms and the Weyl correspondence between $c$-number functions in the phase space and quantum mechanical operators in the Hilbert space so that quantum mechanical composition of functions relies on the star-product.

The Wigner transform of an operator $\hat{A}$ with its matrix elements $\langle\sigma|\hat{A}|\sigma'\rangle = A(\sigma, \sigma')$ in the position space is the function [46–49]

$$A_{\mathrm{W}}(\sigma, p, \hbar) = \int d\sigma' \left\langle \sigma + \frac{\sigma'}{2} \left| \hat{A} \right| \sigma - \frac{\sigma'}{2} \right\rangle e^{-\frac{ip\sigma'}{\hbar}}\,, \tag{14}$$

in the phase space. This maps a quantum mechanical operator $\hat{A}$ in the Hilbert space to a function in the phase space. The inverse operation is the Weyl transform which relates a function $B_{\mathrm{W}}(\sigma, p, \hbar)$ in the phase space to a quantum operator $\hat{B}$ in the Hilbert space with matrix elements

$$\langle\sigma|\hat{B}|\sigma'\rangle = \int \frac{dp}{2\pi\hbar} B_{\mathrm{W}}\left(\frac{\sigma + \sigma'}{2}, p, \hbar\right) e^{\frac{ip(\sigma-\sigma')}{\hbar}}\,. \tag{15}$$

When we deal with more than one particle, we need to include the effects of quantum statistics in the Wigner transform. For the Fermi-gas, the Wigner transform of the $s$-body operator $\mathcal{O}^{(s)}$ is obtained by taking the anti-symmetrized operators $P_{\mathrm{A}}\mathcal{O}^{(s)}$ where $P_{\mathrm{A}}$ is the projection operator

$$P_{\mathrm{A}} = \frac{1}{s!} \sum_{v \in S_N} (-1)^{\epsilon(v)} v\,, \tag{16}$$

which anti-symmetrize the states [44].

The Wigner transform of a product of operators $\hat{A}$ and $\hat{B}$ is given by [46–49]

$$(\hat{A}\hat{B})_{\mathrm{W}} = A_{\mathrm{W}} \star B_{\mathrm{W}}\,. \tag{17}$$

Here $\star$ is the star operation

$$\star = \exp\left[\frac{i\hbar}{2}\left(\overleftarrow{\partial}_\sigma \overrightarrow{\partial}_p - \overleftarrow{\partial}_p \overrightarrow{\partial}_\sigma\right)\right]\,, \tag{18}$$

where the derivatives act on the left or on the right according to the directions indicated by the arrows. One can express all operators of quantum mechanics in terms of the Wigner transforms of operators in such a way that the semi-classical expansion of an operator $\hat{A}$ is given by

$$A_{\mathrm{W}}(\sigma, p, \hbar) = \sum_{n=0}^{\infty} A_n(\sigma, p) \hbar^n, \tag{19}$$

where $A_0$ is the classical limit of $\hat{A}$.

From the Baker-Campbell-Hausdorff formula, (12) and (17) we find the Wigner transform of the Hamiltonian

$$H_{\mathrm{W}}(\sigma, p) = T(p) + U(\sigma) - \frac{\hbar^2}{12}(T'(p))^2 U''(\sigma) + \frac{\hbar^2}{24}(U'(\sigma))^2 T''(p) + \mathcal{O}(\hbar^4). \tag{20}$$

Furthermore, the semi-classical expansion of arbitrary function $f(\hat{H})$ of the Hamiltonian operator $\hat{H}$ is given by

$$f_{\mathrm{W}}(\hat{H})(\sigma, p, \hbar) = \sum_{r=0}^{\infty} \frac{1}{r!} f^{(r)}(H_{\mathrm{W}}(\sigma, p)) \mathcal{G}_r(\sigma, p; \hbar), \tag{21}$$

where $f^{(r)}(a)$ is the $r$-th derivative of $f(x)$ evaluated at $x = a$ and

$$\mathcal{G}_r(\sigma, p; \hbar) = \left[ \left( \hat{H} - H_{\mathrm{W}}(\sigma, p) \right)^r \right]_{\mathrm{W}} (\sigma', p'; \hbar) \Bigg|_{(\sigma', p') = (\sigma, p)}, \tag{22}$$

is the universal coefficients in the expansion around $H_{\mathrm{W}}$. It follows that $\mathcal{G}_r$ is an even function of $\hbar$ such that

$$\mathcal{G}_r(\sigma, p; \hbar) = \mathcal{O}(\hbar^{2(n+1)}), \tag{23}$$

for the largest integer $n < \frac{r}{3}$. For example, we have [50, 51]

$$\begin{aligned}
\mathcal{G}_0 &= 1, \\
\mathcal{G}_1 &= 0, \\
\mathcal{G}_2 &= -\frac{\hbar^2}{4} \left[ \frac{\partial^2 H_{\mathrm{W}}}{\partial \sigma^2} \frac{\partial^2 H_{\mathrm{W}}}{\partial p^2} - \left( \frac{\partial^2 H_{\mathrm{W}}}{\partial \sigma \partial p} \right)^2 \right] + \mathcal{O}(\hbar^4), \\
\mathcal{G}_3 &= -\frac{\hbar^2}{4} \left[ \left( \frac{\partial H_{\mathrm{W}}}{\partial \sigma} \right)^2 \frac{\partial^2 H_{\mathrm{W}}}{\partial p^2} + \left( \frac{\partial H_{\mathrm{W}}}{\partial p} \right)^2 \frac{\partial^2 H_{\mathrm{W}}}{\partial \sigma^2} - 2 \frac{\partial H_{\mathrm{W}}}{\partial \sigma} \frac{\partial H_{\mathrm{W}}}{\partial p} \frac{\partial^2 H_{\mathrm{W}}}{\partial \sigma \partial p} \right] + \mathcal{O}(\hbar^4).
\end{aligned} \tag{24}$$

In particular, the Wigner transform of a distribution operator at zero temperature is given by [50]

$$\theta_{\mathrm{W}} \left( \frac{2\pi\mu}{\epsilon_1} - \hat{H} \right) = \theta \left( \frac{2\pi\mu}{\epsilon_1} - H_{\mathrm{W}}(\sigma, p) \right) + \sum_{r=2}^{\infty} \frac{\mathcal{G}_r}{r!} \delta^{(r-1)} \left( \frac{2\pi\mu}{\epsilon_1} - H_{\mathrm{W}}(\sigma, p) \right), \tag{25}$$

where $\theta(x)$ is the Heaviside step function. By taking the trace of the distribution operator, we get the function $n_{\mathrm{W}}(\mu)$ that counts the number of eigenstates whose energy is less than $\frac{2\pi\mu}{\epsilon_1}$.

In the thermodynamic limit $N \to \infty$, the behavior of the system is semi-classical and the trace can be evaluated as an integral over the phase space. Therefore we obtain

$$n_{\mathrm{W}}(\mu) = \int d\sigma dp \, \theta \left( \frac{2\pi\mu}{\epsilon_1} - H_{\mathrm{W}}(\sigma, p) \right) + \sum_{r=2}^{\infty} \int d\sigma dp \frac{\mathcal{G}_r}{r!} \delta^{(r-1)} \left( \frac{2\pi\mu}{\epsilon_1} - H_{\mathrm{W}}(\sigma, p) \right). \tag{26}$$

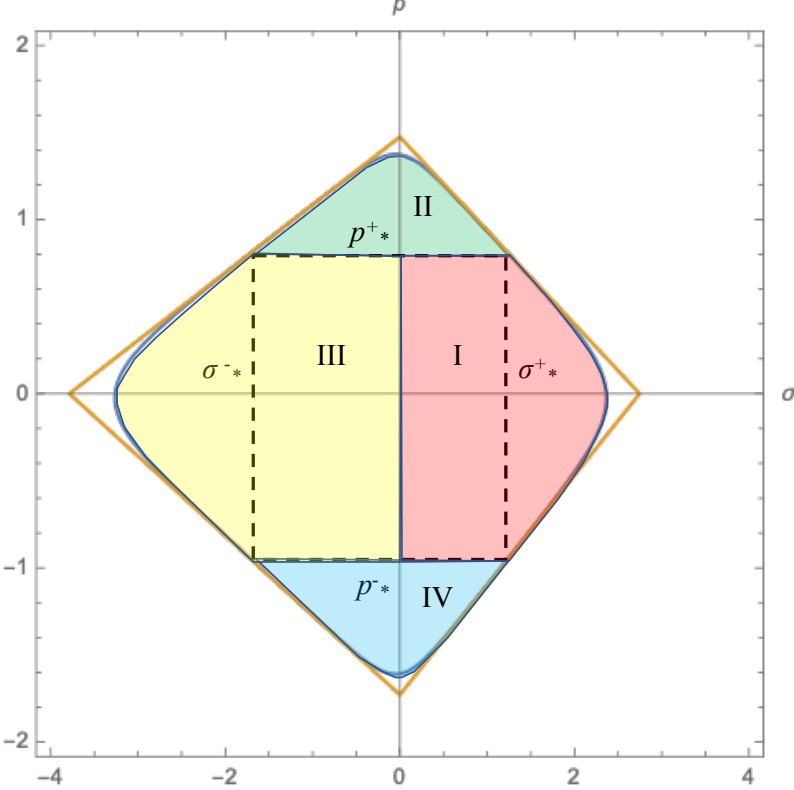

Figure 1: The quantum corrected curve (blue line) and the polygon as its large $N$ approximation (orange line). We divide the Fermi surface into four regions I, II, III and IV.

Here the first term is the area of the quantum corrected Fermi surface defined by the equation

$$H_{\mathrm{W}}(\sigma, p) = \frac{2\pi\mu}{\epsilon_1}, \tag{27}$$

and the second term is the quantum corrections arising from the semi-classical expansion of the distribution operator. Then the density of energy eigenstates is given by

$$\rho_{\mathrm{W}}(\mu) = \frac{dn_{\mathrm{W}}(\mu)}{d\mu}. \tag{28}$$

Let us first evaluate the area of the quantum corrected Fermi surface in the limit $\mu \to \infty$:

$$\mathrm{Vol}(\mu) := \int d\sigma\, dp\, \theta\left(\frac{2\pi\mu}{\epsilon_1} - H_{\mathrm{W}}(\sigma, p)\right). \tag{29}$$

Since we have

$$\log\left(2\cosh\pi x\right) = \pi x + \log(1 + e^{-2\pi x}) = \pi x + \sum_{k=1}^{\infty} (-1)^{k+1} \frac{e^{-2k\pi x}}{k}, \tag{30}$$

the potential term $U(\sigma)$ and its derivative have the asymptotics

$$
U(\sigma) = \begin{cases} \pi(l - 2i\zeta)\sigma + \mathcal{O}(e^{-\sigma}) & \sigma \to \infty \\ -\pi(l + 2i\zeta)\sigma + \mathcal{O}(e^{\sigma}) & \sigma \to -\infty \end{cases},
$$

$$
U'(\sigma) = \begin{cases} \pi(l - 2i\zeta) + \mathcal{O}(e^{-\sigma}) & \sigma \to \infty \\ -\pi(l + 2i\zeta) + \mathcal{O}(e^{\sigma}) & \sigma \to -\infty \end{cases},
$$

$$
U''(\sigma) = \begin{cases} \mathcal{O}(e^{-\sigma}) & \sigma \to \infty \\ \mathcal{O}(e^{\sigma}) & \sigma \to -\infty \end{cases}, \tag{31}
$$

and the kinetic term $T(p)$ and its derivative have the asymptotics

$$
T(p) = \begin{cases} \pi(1 - 2im)p + \mathcal{O}(e^{-p}) & p \to \infty \\ -\pi(l + 2im)p + \mathcal{O}(e^{p}) & p \to -\infty \end{cases},
$$

$$
T'(p) = \begin{cases} \pi(1 - 2im) + \mathcal{O}(e^{-p}) & p \to \infty \\ -\pi(l + 2im) + \mathcal{O}(e^{p}) & p \to -\infty \end{cases},
$$

$$
T''(p) = \begin{cases} \mathcal{O}(e^{-p}) & p \to \infty \\ \mathcal{O}(e^{p}) & p \to -\infty \end{cases}. \tag{32}
$$

Let $(\sigma_*^+, p_*^+)$, $(\sigma_*^+, p_*^-)$, $(\sigma_*^-, p_*^+)$ and $(\sigma_*^-, p_*^-)$ be points in the quantum curve where

$$
p_*^+ = \frac{\mu}{\epsilon_1(1 - 2im)}, \qquad\qquad p_*^- = -\frac{\mu}{\epsilon_1(1 + 2im)}. \tag{33}
$$

It then follows from (27), (31) and (32) that

$$
\sigma_*^+ = \frac{\mu}{\epsilon_1(l - 2i\zeta)} + \mathcal{O}(e^{-\mu}), \qquad\qquad \sigma_*^- = -\frac{\mu}{\epsilon_1(l + 2i\zeta)} + \mathcal{O}(e^{-\mu}), \tag{34}
$$

where the exponentially small corrections in $\mu$ are power series in $\hbar^2$.

We divide the quantum corrected Fermi surface into four domains:

$$
\begin{aligned}
\text{I}:& \quad 0 \leq \sigma, \quad p_*^- \leq p \leq p_*^+, \\
\text{II}:& \quad p_*^+ \leq p, \\
\text{III}:& \quad \sigma \leq 0, \quad p_*^- \leq p \leq p_*^+, \\
\text{IV}:& \quad p \leq p_*^-,
\end{aligned} \tag{35}
$$

as shown in Figure 1. The area is then the sum of these four domains:

$$
\text{Vol}(\mu) = \text{Vol}_{\text{I}} + \text{Vol}_{\text{II}} + \text{Vol}_{\text{III}} + \text{Vol}_{\text{IV}}.
$$

On the quantum curve in the region I and III, the exponential terms in $\sigma$ are larger than those in $\mu$. Thus we have the potential term and its derivatives

$$
U(\sigma) = \begin{cases} \pi(l - 2i\zeta)\sigma + \mathcal{O}(e^{-\sigma}) & \text{for I} \\ -\pi(l + 2i\zeta)\sigma + \mathcal{O}(e^{-\sigma}) & \text{for III} \end{cases},
$$

$$
U'(\sigma) = \begin{cases} \pi(l - 2i\zeta) + \mathcal{O}(e^{-\sigma}) & \text{for I} \\ -\pi(l + 2i\zeta) + \mathcal{O}(e^{-\sigma}) & \text{for III} \end{cases},
$$

$$
U''(\sigma) = \begin{cases} \mathcal{O}(e^{-\sigma}) & \text{for I} \\ \mathcal{O}(e^{-\sigma}) & \text{for III} \end{cases}, \tag{36}
$$

and the Wigner transform of the Hamiltonian

$$H_W(\sigma, p) = \begin{cases} \pi(l - 2i\zeta)\sigma + T(p) + \frac{\hbar^2}{24}\pi^2(l - 2i\zeta)^2 T''(p) + \mathcal{O}(\hbar^4) & \text{for I} \\ -\pi(l + 2i\zeta)\sigma + T(p) + \frac{\hbar^2}{24}\pi^2(l + 2i\zeta)^2 T''(p) + \mathcal{O}(\hbar^4) & \text{for III} \end{cases}. \tag{37}$$

Therefore we can solve $\sigma$ along the quantum curve

$$\sigma = \begin{cases} .\sigma^+(\mu, p) = \frac{1}{\pi(l - 2i\zeta)}\left[\frac{2\pi\mu}{\epsilon_1} - T(p) - \frac{\hbar^2}{24}\pi^2(l - 2i\zeta)^2 T''(p)\right] & \text{for I} \\ \sigma^-(\mu, p) = -\frac{1}{\pi(l + 2i\zeta)}\left[\frac{2\pi\mu}{\epsilon_1} - T(p) - \frac{\hbar^2}{24}\pi^2(l + 2i\zeta)^2 T''(p)\right] & \text{for III} \end{cases}. \tag{38}$$

On the quantum curve in the regions II and IV, the exponential terms in $p$ are larger than those in $\mu$ so that the kinetic term and its derivatives become

$$T(p) = \begin{cases} \pi(1 - 2im)p + \mathcal{O}(e^{-\mu}) & \text{for II} \\ -\pi(1 + 2im)p + \mathcal{O}(e^{-\mu}) & \text{for IV} \end{cases},$$

$$T'(p) = \begin{cases} \pi(1 - 2im) + \mathcal{O}(e^{-\mu}) & \text{for II} \\ -\pi(1 + 2im) + \mathcal{O}(e^{-\mu}) & \text{for IV} \end{cases},$$

$$T''(p) = \begin{cases} \mathcal{O}(e^{-\mu}) & \text{for II} \\ \mathcal{O}(e^{-\mu}) & \text{for IV} \end{cases}, \tag{39}$$

and the Wigner transform of the Hamiltonian reduces to

$$H_W(\sigma, p) = \begin{cases} U(\sigma) + \pi(1 - 2im)p - \frac{\hbar^2}{12}\pi^2(1 - 2im)^2 U''(\sigma) + \mathcal{O}(\hbar^4) & \text{for II} \\ U(\sigma) - \pi(1 + 2im)p - \frac{\hbar^2}{12}\pi^2(1 + 2im)^2 U''(\sigma) + \mathcal{O}(\hbar^4) & \text{for IV} \end{cases}. \tag{40}$$

Thus we can solve for $p$ along the quantum curve

$$p = \begin{cases} p^+(\mu, \sigma) = \frac{1}{\pi(1 - 2im)}\left[\frac{2\pi\mu}{\epsilon_1} - U(\sigma) + \frac{\hbar^2}{12}\pi^2(1 - 2im)^2 U''\right] & \text{for II} \\ p^-(\mu, \sigma) = -\frac{1}{\pi(1 + 2im)}\left[\frac{2\pi\mu}{\epsilon_1} - U(\sigma) + \frac{\hbar^2}{12}\pi^2(1 + 2im)^2 U''\right] & \text{for IV} \end{cases}. \tag{41}$$

Putting the area of the quantum Fermi surface of the region I, II, III and IV together, we finally obtain the quantum corrected area $\mathrm{Vol}(\mu)$ of Fermi surface [3]

$$\begin{aligned} \mathrm{Vol}(\mu) &= \mathrm{Vol_I} + \mathrm{Vol_{II}} + \mathrm{Vol_{III}} + \mathrm{Vol_{IV}} \\ &= -\frac{2l\mu^2}{\epsilon_2(\epsilon_1 + \epsilon_2)(l^2 + 4\zeta^2)} - \frac{l}{6(l^2 + 4\zeta^2)} + \frac{l(\epsilon_1^2 + \epsilon_1\epsilon_2 + \epsilon_2^2)}{24\epsilon_2(\epsilon_1 + \epsilon_2)} \\ &= n_2\mu^2 + n_0. \end{aligned} \tag{42}$$

The quantum corrections that correspond to the second term in (25), which are associated with the semi-classical expansion of a function of the Hamiltonian turn out to yield only the non-perturbative corrections of order $e^{-\mu}$ [30]. Therefore we obtain

$$n_W(\mu) = n_2\mu^2 + n_0 + n_{\mathrm{np}}(\mu), \tag{43}$$

where $n_{\mathrm{np}}(\mu) = \mathcal{O}(\mu e^{-\mu})$ denotes the non-perturbative terms. Since our Hamiltonian is positive, it follows that $n_W(0) = 0$ and $n_{\mathrm{np}}(0) = -n_0$ after resumming all the non-perturbative corrections.

---

[3] see Appendix B.1 for the detail

To obtain the leading and next-to-leadingg coefficients that show up in the free energy, we observe that the grand canonical potential can be expressed in terms of the density (28) of eigenstates:

$$
\begin{aligned}
J(\mu) &= \int_0^\infty d\mu' \rho_W(\mu') \log(1 + e^{\frac{2\pi(\mu - \mu')}{\epsilon_1}}) \\
&\approx 2n_2 \left(\frac{\epsilon_1}{2\pi}\right)^2 \int_0^\infty d\nu \, \nu \log(1 + e^{\frac{2\pi\mu}{\epsilon_1} - \nu}) + \frac{2\pi\mu}{\epsilon_1} \int_0^\infty d\mu' \frac{dn_{np}(\mu')}{d\mu'} \\
&= -2n_2 \left(\frac{\epsilon_1}{2\pi}\right)^2 \mathrm{Li}_3(-e^{\frac{2\pi\mu}{\epsilon_1}}) + n_0 \frac{2\pi\mu}{\epsilon_1} \mu \, .
\end{aligned}
\tag{44}
$$

According to the asymptotics of the trilogarithm

$$
\mathrm{Li}_3(-e^x) = -\frac{x^3}{6} - \frac{\pi^2}{6} x + \mathcal{O}(e^{-x}),
\tag{45}
$$

we get

$$
\begin{aligned}
J(\mu) &= \frac{n_2}{3}\left(\frac{2\pi}{\epsilon_1}\right)\mu^3 + \left[\frac{\pi^2}{3} n_2 \left(\frac{2\pi}{\epsilon_1}\right)^{-1} + n_0 \left(\frac{2\pi}{\epsilon_1}\right)\right] \mu + A + J_{np}(\mu) \\
&= \frac{C}{3}\mu^3 + B\mu + A + J_{np}(\mu),
\end{aligned}
\tag{46}
$$

where

$$
C = \frac{4\pi l}{\epsilon_1 \epsilon_2 \epsilon_3 (l^2 + 4\zeta^2)},
\tag{47}
$$

and

$$
B = -\frac{\pi l(\epsilon_1^2 + \epsilon_2^2 + \epsilon_3^2)(l^2 - 4 + 4\zeta^2)}{24\epsilon_1 \epsilon_2 \epsilon_3 (l^2 + 4\zeta^2)}.
\tag{48}
$$

We see that the leading coefficient (47) and the next-to-leading coefficient (48) are actually invariant under the triality transformation (2). The overall factor $\frac{1}{\epsilon_1 \epsilon_2 \epsilon_3}$ can be interpreted as the equivariant volume of the $\Omega$-deformed planes $\mathbb{C}_{\epsilon_1} \times \mathbb{C}_{\epsilon_2} \times \mathbb{C}_{\epsilon_3}$ in the background (1) of the twisted M-theory.

In the next section, we extend this computation to correlation functions for Coulomb branch operators.

## 3 Coulomb branch correlators

The 3d $\mathcal{N} = 4$ supersymmetric gauge theory generically contains two types of half-BPS local operators, i.e. the Coulomb and Higgs branch operators, which parametrize two branches of supersymmetric vacua, the Coulomb and Higgs branches respectively. The Coulomb branch operators can be built out of the monopole operators $v_{n_*}$ [4]

dressed by the vector multiplet scalar fields $\varphi$. They can be expanded as a sum over the monopole operators

$$
\mathcal{O}_C = \sum_{n_*} R_{n_*}(\varphi, m_{\mathbb{C}}) v_{n_*},
\tag{49}
$$

---

[4]The monopole operators are labeled by the GNO charge $\pm A_a$, $a = 1, \cdots, r$ where $A$ is the cocharacter that specifies an embedding of a $U(1)$ monopole singularity into the gauge group $G$ and $r$ is the rank of $G$. For $G = U(N)$ we have $A = (A_1, \cdots, A_r) \in \mathbb{Z}^r$ and we denote the integer-valued charge by $n_*$.

where $R_{n_*}(\varphi, m_{\mathbb{C}})$ are polynomials in $\varphi$. The sphere correlation function of the Coulomb branch operators for the ADHM theory takes the form[5]

$$\langle \mathcal{O}_C \rangle = \frac{1}{N!} \int \prod_{i=1}^{N} d\sigma_i \, e^{2\pi i \zeta \sigma_i} \frac{\prod_{i<j} 4 \sinh^2 \pi(\sigma_i - \sigma_j)}{\prod_{i,j=1}^{N} 2 \cosh \pi(\sigma_i - \sigma_j - m)(\prod_{i=1}^{N} 2 \cosh \pi \sigma_i)^l} R_0(-i\sigma, -im). \tag{50}$$

The sphere correlation functions of the Coulomb branch operators can be universally expressed in an algebraic way in terms of the twisted traces over the Verma modules of the quantized Coulomb branch algebra [19]. The factor $R_0(-i\sigma, -im)$ inserted in the correlation function is pulled back from generators in the quantized Coulomb branch algebra. Since the non-trivial twisted traces involving the monopole operators or equivalently shift operators can appear only when they simply shift the vector multiplet scalar fields, only some insertion of non-periodic part without the shift, i.e. $R_0(-i\sigma, -im)$ in the integrand will lead to distinct Coulomb branch correlation functions with some change of residues.

## 3.1 Quantized Coulomb branch algebra

There exist two types of $\Omega$-deformations for the 3d $\mathcal{N} = 4$ supersymmetric gauge theory, in which two kinds of non-commutative algebras of the topological Coulomb and Higgs branch operators emerge. They are called the quantized Coulomb and Higgs branch algebras [15,16]. The quantized Coulomb branch algebra $\mathcal{A}^C_{N,l;\epsilon_1,\epsilon_2}$ of the ADHM theory is isomorphic to the spherical part $\mathbf{SH}^{\mathrm{cyc}}_{N,l}$ of the cyclotomic rational Cherednik algebra [42]. The algebra $\mathcal{A}^C_{N,l;\epsilon_1,\epsilon_2}$ can be also identified with the shifted Yangian $Y_l(m_i)$ of $\widehat{\mathfrak{gl}}(1)$ which is obtained by deforming the subalgebra of an affine Yangian $Y(\widehat{\mathfrak{gl}}(1))$ [52].

Let us introduce coordinates $w_a$ and shift operators $v_a$, $v_a^{-1}$, $a = 1, \cdots, N$ which obey

$$\begin{aligned} [w_a, w_b] &= 0, & [v_a, v_b] &= 0, \\ v_a^{-1} v_b &= \delta_{ab}, & v_a v_b^{-1} &= \delta_{ab}, \\ [v_a^{\pm}, w_b] &= \pm \delta_{ab} \epsilon_1 v_a^{\pm}. \end{aligned} \tag{51}$$

The algebra $\mathcal{A}^C_{N,l;\epsilon_1,\epsilon_2}$ is generated by the operator

$$D_{0,n} = \sum_{a=1}^{N} \frac{(-\epsilon_1)^n}{n} \left[ B_n\left(-\frac{w_a}{\epsilon_1}\right) - B_n\left(\frac{(a-1)\epsilon_2}{\epsilon_1}\right) \right], \qquad n \geq 1 \tag{52}$$

where $B_n(x)$ is the Bernoulli polynomial as well as raising and lowering operators which take the forms:

$$e_n = \sum_{a=1}^{N} (w_a + \epsilon_1)^n \prod_{b \neq a} \frac{w_a - w_b - \epsilon_2}{w_a - w_b} \prod_{a=1}^{N} v_a, \tag{53}$$

$$f_{n+l} = \sum_{a=1}^{N} w_a^n \prod_{b \neq a} \frac{w_a - w_b + \epsilon_2}{w_a - w_b} \prod_{a=1}^{N} \left( \prod_{i=1}^{l} (w_a - \epsilon_1 - m_i) v_a^{-1} \right), \tag{54}$$

for non-negative integer $n$. Here $m_i$ are mass parameters for the $SU(l)$ flavor symmetry. They

---

[5]See [5,6] for the result of supersymmetric localization.

obey the relations

$$[D_{0,n}, D_{0,m}] = 0 \,, \tag{55}$$

$$[D_{0,n}, e_m] = -\epsilon_1 e_{n+m-1} \,, \tag{56}$$

$$[D_{0,n}, f_m] = \epsilon_1 f_{n+m-1} \,, \tag{57}$$

$$3[e_2, e_1] - [e_3, e_0] + (\epsilon_1^2 + \epsilon_1\epsilon_2 + \epsilon_2^2)[e_1, e_0] + \epsilon_1\epsilon_2(\epsilon_1 + \epsilon_2)e_0^2 = 0 \,, \tag{58}$$

$$3[f_2, f_1] - [f_3, f_0] + (\epsilon_1^2 + \epsilon_2\epsilon_2 + \epsilon_2^2)[f_1, f_0] - \epsilon_1\epsilon_2(\epsilon_1 + \epsilon_2)f_0^2 = 0 \,, \tag{59}$$

$$[e_0, [e_0, e_1]] = [f_0, [f_0, f_1]] = 0 \,, \tag{60}$$

$$[e_n, f_m] = \epsilon_1 h_{n+m} \,. \tag{61}$$

Here the operator $h_n$ can be determined by the relation

$$1 - \epsilon_2(\epsilon_1 + \epsilon_2) \sum_{n \geq 0} h_n z^{n+1}$$
$$= \prod_{i=1}^{l} (1 - (m_i + \epsilon_1)z) \frac{(1 - (\epsilon_1 + \epsilon_2)z)(1 + N\epsilon_2 z)}{1 - (\epsilon_1 + (1-N)\epsilon_2)z} \exp\left[ -\sum_{n \geq 0} \frac{D_{0,n+1}\varphi_n(z)}{\epsilon_1} \right] \,, \tag{62}$$

where

$$\varphi_n(z) = z^n \Big[ G_n(1 + \epsilon_1 z) - G_n(1 - \epsilon_1 z) + G_n(1 + \epsilon_2 z) - G_n(1 - \epsilon_2 z)$$
$$+ G_n(1 - (\epsilon_1 + \epsilon_2)z) - G_n(1 + (\epsilon_1 + \epsilon_2)z) \Big] \,, \tag{63}$$

$$G_n = \begin{cases} -\log z & \text{for } n = 0 \\ \frac{z^{-n} - 1}{n} & \text{for } n \geq 1 \end{cases} \,. \tag{64}$$

## 3.2 The $d_n$ operators

There is an alternative presentation of the algebra in such a way that all generators can take the form of $\epsilon_1$ times a triality-invariant expression [38]. We can introduce the Hamiltonian operator

$$W[f] = \sum_{a=1}^{N} f(w_a) \,, \tag{65}$$

associated to any polynomial $f(w)$ in $w$.

When we choose polynomials

$$p_n(\sigma) = (-1)^n \epsilon_1^{n-1} B_n\left(\frac{1}{2} - i\sigma\right)$$
$$= i^n \epsilon_1^{n-1} \sum_{\substack{k:\text{even} \\ 0 \leq k \leq n}} (-1)^{\frac{k}{2}+1} \frac{(2^k - 2)B_k}{2^k k!} \frac{n!}{(n-k)!} \sigma^{n-k} \,, \tag{66}$$

where $B_n(x)$ is the Bernoulli polynomial, which satisfy the recursion relation [6]

$$p_n\left(w - \frac{i}{2}\right) - p_n\left(w + \frac{i}{2}\right) = n(i\epsilon_1 w)^{n-1} \,, \tag{67}$$

we obtain the operator

$$d_n = W[p_n] \,. \tag{68}$$

---

[6]This takes a similar form as the recursion relation for the Bernoulli polynomial $B_n(x+1) - B_n(x) = nx^{n-1}$.

It has a generating function

$$\frac{1}{\epsilon_1^2}\psi'\left(\frac{1}{2}-iw+\frac{z}{\epsilon_1}\right)=\sum_n\frac{p_n(w)}{z^{n+1}}, \tag{69}$$

where $\psi(z)$ is the digamma function.

For example, we have

$$p_1(w)=iw, \tag{70}$$

$$p_2(w)=-\epsilon_1\left(w^2+\frac{1}{12}\right), \tag{71}$$

$$p_3(w)=-i\epsilon_1^2\left(w^3+\frac{w}{4}\right), \tag{72}$$

$$p_4(w)=\epsilon_1^3\left(w^4+\frac{1}{2}w^2+\frac{7}{240}\right), \tag{73}$$

$$p_5(w)=i\epsilon_1^4\left(w^5+\frac{5}{6}w^3+\frac{7}{48}w\right), \tag{74}$$

$$p_6(w)=-\epsilon_1^5\left(w^6+\frac{5}{4}w^4+\frac{7}{16}w^2+\frac{31}{1344}\right). \tag{75}$$

Making use of the operator $d_n$ given by (68), one can also build the other generators in the quantized Coulomb branch algebra $\mathcal{A}_{N,l;\epsilon_1,\epsilon_2}^C$ which take triality-invariant fashion up to the overall $\epsilon_1$ factor, as discussed in [38]. It manifests the symmetry of the algebra under the triality symmetry (2).

## 3.3 Fermi-gas formulation

In terms of the Fermi-gas formulation, we can also evaluate correlation functions of the Coulomb branch operators. The treatment is very similar to the previous work [53] for Wilson loop correlators in ABJM theory. We can rewrite the sphere correlation function (50) as

$$\langle\mathcal{O}_C\rangle=\frac{1}{N!}\sum_{\nu\in S_N}(-1)^{\epsilon(\nu)}\int\prod_{i=1}^N d\sigma_i\prod_{i=1}^N\rho(\sigma_i,\sigma_{\nu(i)})R_0(-i\sigma,-im). \tag{76}$$

First, we consider the sphere one-point function of a positive power function:

$$\langle\sigma^n\rangle:=\frac{1}{N!}\sum_{\nu\in S_N}(-1)^{\epsilon(\nu)}\int\prod_{i=1}^N d\sigma_i\prod_{i=1}^N\rho(\sigma_i,\sigma_{\nu(i)})\left(\sum_{i=1}^N\sigma_i\right)^n. \tag{77}$$

As we will see later, this is an important building block to compute the thermodynamic limit $N\to\infty$ of the one-point function $\langle d_n\rangle$. To study the large $N$ limit of the one-point function (77), we evaluate an average of the one-point function by integrating over the phase space in terms of the Wigner transform of the distribution operator (25):

$$n_W^{\sigma^n}(\mu):=\int d\sigma dp\,\theta_W\left(\frac{2\pi\mu}{\epsilon_1}-H_W(\sigma,p)\right)\sigma^n \tag{78}$$

$$=\int d\sigma dp\,\theta\left(\frac{2\pi\mu}{\epsilon_1}-H_W(\sigma,p)\right)\sigma^n$$

$$+\sum_{r=2}^\infty\int d\sigma dp\,\frac{G_r}{r!}\delta^{(r-1)}\left(\frac{2\pi\mu}{\epsilon_1}-H_W(\sigma,p)\right)\sigma^n, \tag{79}$$

where the second line involves the corrections from the quantum Fermi surface and the third is associated to the corrections from the semi-classical expansion of the distribution operator.

The corrections from the quantum Fermi surface can be evaluated in the same manner in the previous section. Collecting the pieces (170), (171) (175) and (176), we obtain the leading and next-to-leading terms of the average over the quantum Fermi surface in the first line of (78):

$$
\int d\sigma dp \, \theta\left(\frac{2\pi\mu}{\epsilon_1} - H_{\mathrm{W}}(\sigma, p)\right)\sigma^n = \mathrm{Vol}_{\mathrm{I}}^{\sigma^n} + \mathrm{Vol}_{\mathrm{II}}^{\sigma^n} + \mathrm{Vol}_{\mathrm{III}}^{\sigma^n} + \mathrm{Vol}_{\mathrm{IV}}^{\sigma^n}
$$

$$
= -\frac{2\left[(-2)^n(l - 2i\zeta)^{n+1} + 2^n(l + 2i\zeta)^{n+1}\right]}{\epsilon_1^n \epsilon_2(\epsilon_1 + \epsilon_2)(n+1)(n+2)(l^2 + 4\zeta^2)^{n+1}}\mu^{n+2}
$$

$$
- \frac{2^{n-4}}{3\epsilon_1^n}\left[\frac{4 + (l - 2i\zeta)^2}{(l - 2i\zeta)^{n+1}} + (-1)^n\frac{4 + (l + 2i\zeta)^2}{(l + 2i\zeta)^{n+1}}\right]\mu^n. \tag{80}
$$

Next proceed to the $\mathcal{G}_r$ corrections arising from the semi-classical expansion of the distribution operator. To order $\hbar^2$ corrections only come from $\mathcal{G}_2$ and $\mathcal{G}_3$ in (24):

$$
\frac{1}{2}\int d\sigma dp \, \mathcal{G}_2\left[\frac{\partial}{\partial\left(\frac{2\pi\mu}{\epsilon_1}\right)}\delta\left(\frac{2\pi\mu}{\epsilon_1} - H_{\mathrm{W}}(\sigma, p)\right)\right]\sigma^n
$$

$$
+ \frac{1}{6}\int d\sigma dp \, \mathcal{G}_3\left[\frac{\partial^2}{\partial\left(\frac{2\pi\mu}{\epsilon_1}\right)^2}\delta\left(\frac{2\pi\mu}{\epsilon_1} - H_{\mathrm{W}}(\sigma, p)\right)\right]\sigma^n. \tag{81}
$$

Since we have

$$
\delta\left(\frac{2\pi\mu}{\epsilon_1} - H_{\mathrm{W}}(\sigma, p)\right) = \frac{\delta(\sigma - \sigma^-(\mu, p))}{\left|\frac{\partial H_{\mathrm{W}}(\sigma, p)}{\partial\sigma}\right|} + \frac{\delta(\sigma - \sigma^+(\mu, p))}{\left|\frac{\partial H_{\mathrm{W}}(\sigma, p)}{\partial\sigma}\right|}
$$

$$
= \frac{\delta(p - p^-(\mu, \sigma))}{\left|\frac{\partial H_{\mathrm{W}}(\sigma, p)}{\partial p}\right|} + \frac{\delta(p - p^+(\mu, \sigma))}{\left|\frac{\partial H_{\mathrm{W}}(\sigma, p)}{\partial p}\right|}, \tag{82}
$$

$\mathcal{G}_2$ and $\mathcal{G}_3$ in (81) are evaluated along the quantum curves (38) and (41)

$$
\mathcal{G}_2|_{\sigma=\sigma^\pm(\mu,p)} = 0, \qquad\qquad \mathcal{G}_2|_{p=p^\pm(\mu,\sigma)} = 0,
$$

$$
\mathcal{G}_3|_{\sigma=\sigma^\pm(\mu,p)} = -\frac{\hbar^2\pi^2}{4}(l \mp 2i\zeta)^2 T''(p), \quad \mathcal{G}_3|_{p=p^\pm(\mu,\sigma)} = -\frac{\hbar^2\pi^2}{4}(1 \mp 2im)^2 U''(\sigma). \tag{83}
$$

Consequently, only non-trivial corrections may come from $\mathcal{G}_3$. We find

$$
\frac{1}{6}\int d\sigma dp \, \mathcal{G}_3\left[\frac{\partial^2}{\partial\left(\frac{2\pi\mu}{\epsilon_1}\right)^2}\delta\left(\frac{2\pi\mu}{\epsilon_1} - H_{\mathrm{W}}(\sigma, p)\right)\right]\sigma^n =
$$

$$
= -\frac{\hbar^2}{24}\frac{\partial^2}{\partial\left(\frac{2\pi\mu}{\epsilon_1}\right)^2}\int_{p_*^-}^{p_*^+} dp \, T''(p)\frac{1}{\pi^{n-1}(l - 2i\zeta)^{n-1}}\left[\frac{2\pi\mu}{\epsilon_1} - T(p) - \frac{\hbar^2\pi^2}{24}(l - 2i\zeta)^2 T''(p)\right]^n
$$

$$
- \frac{\hbar^2}{24}\frac{\partial^2}{\partial\left(\frac{2\pi\mu}{\epsilon_1}\right)^2}\int_{p_*^-}^{p_*^+} dp \, T''(p)\frac{(-1)^n}{\pi^{n-1}(l + 2i\zeta)^{n-1}}\left[\frac{2\pi\mu}{\epsilon_1} - T(p) - \frac{\hbar^2\pi^2}{24}(l + 2i\zeta)^2 T''(p)\right]^n
$$

$$
- \frac{\hbar^2}{24}\frac{\partial^2}{\partial\left(\frac{2\pi\mu}{\epsilon_1}\right)^2}\int_{\sigma_*^-}^{\sigma_*^+} d\sigma \, \pi(1 - 2im)U''(\sigma)\sigma^n - \frac{\hbar^2}{24}\frac{\partial^2}{\partial\left(\frac{2\pi\mu}{\epsilon_1}\right)^2}\int_{\sigma_*^-}^{\sigma_*^+} d\sigma \, \pi(1 + 2im)U''(\sigma)\sigma^n, \tag{84}
$$

which involve $\mu^{n-2}$ and lower order terms. Hence the quantum corrections associated to the semi-classical expansion do not contribute to the leading and next-to-leading terms.

Putting all together, we finally arrive at

$$n_{\text{W}}^{\sigma^n}(\mu) = -\frac{2\left[(-2)^n(l-2i\zeta)^{n+1} + 2^n(l+2i\zeta)^{n+1}\right]}{\epsilon_1^n\epsilon_2(\epsilon_1+\epsilon_2)(n+1)(n+2)(l^2+4\zeta^2)^{n+1}}\mu^{n+2}$$
$$- \frac{2^{n-4}}{3\epsilon_1^n}\left[\frac{4+(l-2i\zeta)^2}{(l-2i\zeta)^{n+1}} + (-1)^n\frac{4+(l+2i\zeta)^2}{(l+2i\zeta)^{n+1}}\right]\mu^n. \tag{85}$$

For example, for $l=1$ and $\zeta=0$ the expression reduces to a relatively simple form

$$n_{\text{W}}^{\sigma^n}(\mu) = -\frac{2^{n+1}(1+(-1)^n)}{\epsilon_1^n\epsilon_2(\epsilon_1+\epsilon_2)(n+1)(n+2)}\mu^{n+2} - \frac{5\cdot2^{n-4}(1+(-1)^n)}{3\epsilon_1^n}\mu^n. \tag{86}$$

## 3.4 Grand canonical one-point functions

In this and the next subsections, we would like to evaluate the large $N$ limit of correlation functions of the operators $d_n$. The $k$-point function of $d_n$ generically takes the form

$$\langle d_{n_1} d_{n_2} \cdots d_{n_k} \rangle = \frac{1}{N!}\sum_{\nu\in S_N}(-1)^{\epsilon(\nu)}\int\prod_{i=1}^{N}d\sigma_i\prod_{i=1}^{N}\rho(\sigma_i,\sigma_{\nu(i)})\prod_{j=1}^{k}\left(\sum_i p_{n_j}(\sigma_i)\right). \tag{87}$$

We go to the grand canonical ensemble, and study the large $\mu$ limit, as was done for the partition function. We can easily translate obtained results in the grand canonical ensemble into those in the canonical ensemble.

Let us consider the one-point function. According to the formula (66), the polynomial $p_n(\sigma)$ has the leading and next-to-leading terms:

$$p_n(\sigma) = i^n\epsilon_1^{n-1}\sigma^n + \frac{i^n\epsilon_1^{n-1}n(n-1)}{24}\sigma^{n-2} + \cdots. \tag{88}$$

Thus the leading and next-to-leading terms of the one-point function in the thermodynamic limit can be obtained from (85)

$$n_{\text{W}}^{p_n}(\mu) = i^n\epsilon_1^{n-1}n_{\text{W}}^{\sigma^n}(\mu) + \frac{i^n\epsilon_1^{n-1}n(n-1)}{24}n_{\text{W}}^{\sigma^{n-2}}(\mu)$$
$$= -\frac{i^n 2\left[(-2)^n(l-2i\zeta)^{n+1} + 2^n(l+2i\zeta)^{n+1}\right]}{\epsilon_1\epsilon_2(\epsilon_1+\epsilon_2)(n+1)(n+2)(l^2+4\zeta^2)^{n+1}}\mu^{n+2}$$
$$- \frac{i^n 2^{n-4}\left[\frac{4+(l-2i\zeta)^2}{(l-2i\zeta)^{n+1}} + (-1)^n\frac{4+(l+2i\zeta)^2}{(l+2i\zeta)^{n+1}}\right]}{3\epsilon_1}\mu^n$$
$$- \frac{i^n 2^{n-4}\epsilon_1\left[(-1)^n(l-2i\zeta)^{n-1} + (l+2i\zeta)^{n-1}\right]}{3\epsilon_2(\epsilon_1+\epsilon_2)(l^2+4\zeta^2)^{n-1}}\mu^n$$
$$= c_{n+2}\mu^{n+2} + c_n\mu^n. \tag{89}$$

Making use of the Sommerfeld expansion

$$\frac{1}{1+e^{\beta\hat{H}-\mu}} = \frac{\pi}{\beta}\partial_\mu\csc\left(\frac{\pi}{\beta}\partial_\mu\right)\theta(\mu-\hat{H}), \tag{90}$$

one can express the one-point function of operator $\mathcal{O}$ in the grand canonical ensemble for an ideal Fermi-gas system as [54, 55]

$$
\begin{aligned}
\frac{\langle \mathcal{O} \rangle^{\text{GC}}}{\Xi} &= \text{Tr}\left( \frac{\mathcal{O}}{1 + e^{\beta \hat{H} - \mu}} \right) \\
&= \frac{\pi}{\beta} \partial_\mu \csc\left( \frac{\pi}{\beta} \partial_\mu \right) n_{\text{W}}^{\mathcal{O}}(\mu) \\
&= \left( 1 + \frac{\pi^2}{6\beta^2} \partial_\mu^2 + \frac{7\pi^4}{360\beta^4} \partial_\mu^4 + \cdots \right) n_{\text{W}}^{\mathcal{O}}(\mu),
\end{aligned}
\tag{91}
$$

where

$$
\Xi(\mu) = e^{J(\mu)}, \qquad \langle \mathcal{O} \rangle^{\text{GC}}(\mu) = \sum_{N=1}^{\infty} e^{\frac{2\pi\mu}{\epsilon_1} N} \langle \mathcal{O} \rangle(N).
\tag{92}
$$

By taking $\mathcal{O} = p_n(\sigma)$ and $\beta = 2\pi/\epsilon_1$ in (91), we get from (89) the leading and next-to-leading terms of the grand canonical one-point function of the operator $d_n$:

$$
\begin{aligned}
\frac{\langle d_n \rangle^{\text{GC}}}{\Xi} &= c_{n+2}\mu^{n+2} + \left( \frac{\pi^2}{6}(n+2)(n+1)\left(\frac{2\pi}{\epsilon_1}\right)^{-2} c_{n+2} + c_n \right)\mu^n \\
&= \frac{i^n 2^{n+1}\left[ (-1)^n(l - 2i\zeta)^{n+1} + (l + 2i\zeta)^{n+1} \right]}{\epsilon_1 \epsilon_2 \epsilon_3 (n+1)(n+2)(l^2 + 4\zeta^2)^{n+1}} \mu^{n+2} \\
&\quad + \frac{i^n 2^{n-5}\left[ \frac{4+(l-2i\zeta)^2}{(l-2i\zeta)^{n+1}} + (-1)^n \frac{4+(l+2i\zeta)^2}{(l+2i\zeta)^{n+1}} \right](\epsilon_1^2 + \epsilon_2^2 + \epsilon_3^2)}{3\epsilon_1 \epsilon_2 \epsilon_3} \mu^n + \mathcal{O}(\mu^{n-1}).
\end{aligned}
\tag{93}
$$

We see that the leading and next-to-leading coefficients of the grand canonical one-point functions (93) are exactly invariant under the triality symmetry (2)!

Indeed, it is simple to check that our analytic formula (93) of the grand canonical one-point function reproduces the numerical results in [43] when we specialize $l = 1$ and $\zeta = 0$. We first encode the $\mu$ dependence into $d_0$ by replacing $\mu$ with $\frac{\tau_0}{2\pi}$. Then we can obtain the perturbative correlation functions in [43] by taking the derivatives with respect to $\tau_0$ and setting $\tau_0$ to zero.

For example, the grand canonical one-point function of $d_2$ for $l = 1$ and $\zeta = 0$ is given by

$$
\begin{aligned}
\frac{\langle d_2 \rangle^{\text{GC}}}{\Xi} &= -\frac{4}{3\epsilon_1 \epsilon_2 \epsilon_3}\mu^4 - \frac{5(\epsilon_1^2 + \epsilon_2^2 + \epsilon_3^2)}{12\epsilon_1 \epsilon_2 \epsilon_3}\mu^2 \\
&= -\frac{\tau_0^4}{12\pi^4 \epsilon_1 \epsilon_2 \epsilon_3} - \frac{5(\epsilon_1^2 + \epsilon_2^2 + \epsilon_3^2)\tau_0^2}{48\pi^2 \epsilon_1 \epsilon_2 \epsilon_3}.
\end{aligned}
\tag{94}
$$

The derivatives of (94) with respect to $\tau_0$ lead to

$$
\frac{\partial^4}{\partial \tau_0^4} \frac{\langle d_2 \rangle^{\text{GC}}}{\Xi}\bigg|_{\tau_0 = 0} = -\frac{2}{\pi^4 \sigma_3} = \langle d_2 d_0 d_0 d_0 d_0 \rangle_c^{\text{pert}},
\tag{95}
$$

$$
\frac{\partial^2}{\partial \tau_0^2} \frac{\langle d_2 \rangle^{\text{GC}}}{\Xi}\bigg|_{\tau_0 = 0} = -\frac{5\sigma_2}{12\pi^2 \sigma_3} = \langle d_2 d_0 d_0 \rangle_c^{\text{pert}},
\tag{96}
$$

where

$$
\sigma_2 = \frac{1}{2}(\epsilon_1^2 + \epsilon_2^2 + \epsilon_3^2) = (\epsilon_1^2 + \epsilon_1 \epsilon_2 + \epsilon_2^2),
\tag{97}
$$

$$
\sigma_3 = \epsilon_1 \epsilon_2 \epsilon_3.
\tag{98}
$$

The results (95) and (96) perfectly match with the numerical results in [43].[7]

---

[7] See equation (2.48) in [43].

### 3.5 Grand canonical higher-point functions

Higher-point functions can be evaluated by taking the averages of many-body operators in the ideal Fermi-gas. The analysis is more involved than the one-point function.

Consider a system of $N$ particles whose density matrix is $\rho(\sigma_1,\cdots,\sigma_N;\sigma'_1,\cdots,\sigma'_N)$. The reduced $s$-particle density matrices are defined by [54, 56, 57]

$$\rho_s(\sigma_1,\cdots,\sigma_s;\sigma'_1,\cdots,\sigma'_s;N) = \frac{N!}{(N-s)!}\int d\sigma_{s+1}\cdots d\sigma_N \rho(\sigma_1,\cdots,\sigma_N;\sigma'_1,\cdots,\sigma'_N). \quad (99)$$

The thermal average of an $s$-body operator $\mathcal{O}^{(s)}$ in the canonical ensemble can be calculated in terms of the reduced density matrix (99) as

$$\langle\mathcal{O}^{(s)}\rangle(N) = \frac{1}{s!}\int d\sigma_1\cdots d\sigma_s \mathcal{O}^{(s)}(\sigma_1,\cdots,\sigma_s;N)\rho_s(\sigma_1,\cdots,\sigma_s;\sigma'_1,\cdots,\sigma'_s;N). \quad (100)$$

In the grand canonical ensemble, the reduced density matrix is defined by

$$\rho_s^{\mathrm{GC}}(\sigma_1,\cdots,\sigma_s;\sigma'_1,\cdots,\sigma'_s;z) = \sum_{N=s}^{\infty} z^N \rho_s(\sigma_1,\cdots,\sigma_s;\sigma'_1,\cdots,\sigma'_s;N). \quad (101)$$

For an ideal Fermi-gas, the grand canonical reduced density matrix is given by [54, 55]

$$\rho_s^{\mathrm{GC}}(\sigma_1,\cdots,\sigma_s;\sigma'_1,\cdots,\sigma'_s;z) = \Xi\sum_{v\in S_s}(-1)^{\epsilon(v)}\prod_{i=1}^{s}\left\langle\sigma_i\left|\frac{1}{1+z^{-1}e^{\beta\hat{H}}}\right|\sigma_{v(i)}\right\rangle. \quad (102)$$

The semi-classical average of an $s$-body operator $\mathcal{O}^{(s)}$ for the Fermi-gas in the grand canonical ensemble takes the form

$$\frac{\langle\mathcal{O}^{(s)}\rangle^{\mathrm{GC}}}{\Xi} = \frac{\mathrm{Tr}\langle\rho_s^{\mathrm{GC}}(\mathcal{O}^{(s)})_{\mathrm{W}}\rangle}{\Xi} = \int\prod_{i=1}^{s}d\sigma_i dp_i(P_{\mathrm{A}}\mathcal{O}^{(s)})_{\mathrm{W}}\prod_{i=1}^{s}(\rho_s^{\mathrm{GC}})_{\mathrm{W}}(\sigma_i,p_i), \quad (103)$$

where the trace have been performed by the phase space integration and $P_A$ is the projection operator defined in (16).

The grand canonical $k$-point function can be computed from the average of $s(\leq k)$-body operators in the Fermi-gas.

#### 3.5.1 Two-point functions

Let us see the grand canonical two-point functions of the operator $d_n$. It has contributions from the following one- and two-body operators:

$$\mathcal{O}^{(1)}[d_n^2] = \left(\sum_{i=1}^{N}p_n(\sigma_i)\right)^2, \qquad \mathcal{O}^{(2)}[d_n^2] = \sum_{i\neq j}p_n(\sigma_i)p_n(\sigma_j). \quad (104)$$

From (14) we get the leading and next-to-leading terms of the Wigner transforms of the anti-symmetrized operators for (104)

$$(\mathcal{O}^{(1)}[d_n^2])_{\mathrm{W}} = p_n(\sigma)^2, \quad (105)$$

$$(P_{\mathrm{A}}\mathcal{O}^{(2)}[d_n^2])_{\mathrm{W}} = p_n(\sigma_1)p_n(\sigma_2) - \delta(\sigma_1-\sigma_2)\int dy\, p_n(\sigma_1-\frac{y_1}{2})p_n(\sigma_1+\frac{y_1}{2})e^{\frac{i(p_1-p_2)y}{\hbar}}$$

$$= p_n(\sigma_1)p_n(\sigma_2) - 2\pi\hbar\delta(\sigma_1-\sigma_2)\delta(p_1-p_2)p_n(\sigma_1)^2$$

$$- 2\pi\hbar^3\delta(\sigma_1-\sigma_2)\delta''(p_1-p_2)\frac{n}{4}\sigma_1^{2n-2} + \cdots, \quad (106)$$

where the ellipsis indicates the terms at low orders in $\sigma$ which do not contribute to the leading and next-to-leading coefficinets of the correlation functions.

Plugging the Wigner transforms (105) and (106) into (103), we find the leading and next-to-leading terms of the two-point function

$$
\begin{aligned}
\frac{\langle d_n d_n \rangle^{\mathrm{GC}}}{\Xi} &= \int d\sigma dp (\mathcal{O}^{(1)}[d_n^2])_{\mathrm{W}} \rho_{\mathrm{W}}^{\mathrm{GC}}(\sigma, p) \\
&\quad + \int d\sigma_1 d\sigma_2 dp_1 dp_2 (P_{\mathrm{A}}\mathcal{O}^{(2)}[d_n^2])_{\mathrm{W}} \rho_{\mathrm{W}}^{\mathrm{GC}}(\sigma_1, p_1) \rho_{\mathrm{W}}^{\mathrm{GC}}(\sigma_2, p_2) \\
&= \int d\sigma dp \, p_n(\sigma)^2 \rho_{\mathrm{W}}^{\mathrm{GC}}(1 - \rho_{\mathrm{W}}^{\mathrm{GC}}) + \left( \frac{\langle d_n \rangle^{\mathrm{GC}}}{\Xi} \right)^2 \\
&\quad - 2\pi\hbar^3 (-1)^n \epsilon_1^{2n-2} \int d\sigma dp \frac{n}{2} \sigma^{2n-2} \left( \rho_{\mathrm{W}}^{\mathrm{GC}} \partial_p^2 \rho_{\mathrm{W}}^{\mathrm{GC}} - (\partial_p \rho_{\mathrm{W}}^{\mathrm{GC}})^2 \right).
\end{aligned}
\tag{107}
$$

In the second equality we have combined the average of one-body operator $p_n(\sigma)^2$ with the average of the Wigner transform of the antisymmetrized two-body operator $-2\pi\hbar\delta(\sigma_1 - \sigma_2)\delta(p_1 - p_2)p_n(\sigma_1)^2$ where $\hbar = \frac{1}{2\pi}$ so that they can be evaluated as the one-body integral involving $\rho_{\mathrm{W}}^{\mathrm{GC}}(1 - \rho_{\mathrm{W}}^{\mathrm{GC}})$.

The grand canonical connected two-point function of the operator $d_n$ can be obtained by subtracting the square of the grand canonical one-point functions. Thus we get

$$
\begin{aligned}
\langle d_n d_n \rangle_c^{\mathrm{GC}} &= \frac{\langle d_n d_n \rangle^{\mathrm{GC}}}{\Xi} - \left( \frac{\langle d_n \rangle^{\mathrm{GC}}}{\Xi} \right)^2 \\
&= (-1)^n \epsilon_1^{2n-2} \frac{1}{\beta} \partial_\mu \int d\sigma dp \left( \sigma^{2n} + \frac{n(n-1)}{12} \sigma^{2n-2} + \cdots \right) \rho_{\mathrm{W}}^{\mathrm{GC}} \\
&\quad - 2\pi\hbar^3 (-1)^n \epsilon_1^{2n-2} \int d\sigma dp \frac{n}{2} \sigma^{2n-2} \left( \rho_{\mathrm{W}}^{\mathrm{GC}} \partial_p^2 \rho_{\mathrm{W}}^{\mathrm{GC}} - (\partial_p \rho_{\mathrm{W}}^{\mathrm{GC}})^2 \right) \\
&= (-1)^n \epsilon_1^{2n-2} \left( \frac{1}{\beta} \partial_\mu + \frac{\pi^2}{6\beta^3} \partial_\mu^3 + \cdots \right) \left( \langle \sigma^{2n} \rangle_{\mathrm{W}} + \frac{n(n-1)}{12} \langle \sigma^{2n-2} \rangle_{\mathrm{W}} + \cdots \right) \\
&\quad - 2\pi\hbar^3 (-1)^n \epsilon_1^{2n-2} \int d\sigma dp \frac{n}{2} \sigma^{2n-2} \left( \rho_{\mathrm{W}}^{\mathrm{GC}} \partial_p^2 \rho_{\mathrm{W}}^{\mathrm{GC}} - (\partial_p \rho_{\mathrm{W}}^{\mathrm{GC}})^2 \right),
\end{aligned}
\tag{108}
$$

where we have used the relation

$$
\frac{1}{\beta} \partial_\mu \rho_{\mathrm{W}}^{\mathrm{GC}} = \rho_{\mathrm{W}}^{\mathrm{GC}}(1 - \rho_{\mathrm{W}}^{\mathrm{GC}}),
\tag{109}
$$

in the second equality. We obtain from (108) and (85) the leading term in the connected two-point function:

$$
\langle d_n d_n \rangle_c^{\mathrm{GC}} = \frac{(-1)^n 2^{2n}}{\pi} \frac{\left[ (l - 2i\zeta)^{2n+1} + (l + 2i\zeta)^{2n+1} \right]}{\epsilon_1 \epsilon_2 \epsilon_3 (2n+1)(l^2 + 4\zeta^2)^{2n+1}} \mu^{2n+1} + \mathcal{O}(\mu^{2n-1}).
\tag{110}
$$

In fact, this is invariant under the triality symmetry (2)!

The subleading terms also have contributions from the last term in (108). Unfortunately it seems difficult to evaluate it analytically because of the derivatives of $\rho_{\mathrm{W}}^{\mathrm{GC}}$ in the integrands. However, we can guess a consistent next-to-leading term by requiring the triality invariance,

$$
\begin{aligned}
\Bigg\{ &(-1)^n 2^{2n-4}(2n) \frac{\left[ (l - 2i\zeta)^{2n+1} + (l + 2i\zeta)^{2n+1} \right]}{3\pi \epsilon_1 \epsilon_2 \epsilon_3 (l^2 + 4\zeta^2)^{2n+1}} \\
&+ (-1)^n 2^{2n-5} n(n-1) \frac{\left[ (l - 2i\zeta)^{2n-1} + (l + 2i\zeta)^{2n-1} \right]}{3\pi \epsilon_1 \epsilon_2 \epsilon_3 (2n-1)(l^2 + 4\zeta^2)^{2n-1}} \Bigg\} (\epsilon_1^2 + \epsilon_2^2 + \epsilon_3^2) \mu^{2n-1}.
\end{aligned}
\tag{111}
$$

We will check the validity of this guess below.

It is straightforward to generalize the results (110) and (111) to the connected two-point functions for two distinct operators $d_{n_1}$ and $d_{n_2}$ with $n_1 \neq n_2$ by following the same argument. To make a result simpler, we introduce $N_2 = n_1 + n_2$. The result is

$$
\begin{aligned}
\langle d_{n_1} d_{n_2} \rangle_c^{\mathrm{GC}} = {}& i^{N_2} 2^{N_2} \frac{\left[ (-1)^{N_2}(l-2i\zeta)^{N_2+1} + (l+2i\zeta)^{N_2+1} \right]}{\epsilon_1 \epsilon_2 \epsilon_3 \pi (N_2+1)(l^2+4\zeta^2)^{N_2+1}} \mu^{N_2+1} \\
& + \Bigg\{ i^{N_2} 2^{N_2-4} N_2 \frac{\left[ (-1)^{N_2}(l-2i\zeta)^{N_2+1} + (l+2i\zeta)^{N_2+1} \right]}{3 \epsilon_1 \epsilon_2 \epsilon_3 \pi (l^2+4\zeta^2)^{N_2+1}} \\
& + i^{N_2} 2^{N_2-6} \left[ n_1(n_1-1) + n_2(n_2-1) \right] \frac{\left[ (-1)^{N_2}(l-2i\zeta)^{N_2-1} + (l+2i\zeta)^{N_2-1} \right]}{3 \epsilon_1 \epsilon_2 \epsilon_3 \pi (N_2-1)(l^2+4\zeta^2)^{N_2-1}} \Bigg\} \\
& \times (\epsilon_1^2 + \epsilon_2^2 + \epsilon_3^2) \mu^{N_2-1} + \mathcal{O}(\mu^{N_2-2}).
\end{aligned}
\tag{112}
$$

We should say again that the next-to-leading term in this expression is a guess based on the triality.

Now we check that our analytic formula (112) of the grand canonical connected two-point function reproduces the numerical results in [43] as special cases with $l=1$ and $\zeta=0$. For example, in a similar manner for the one-point function, we have

$$
\langle d_1 d_1 \rangle_c^{\mathrm{GC}} = -\frac{8}{3\epsilon_1\epsilon_2\epsilon_3\pi}\mu^3 - \frac{\epsilon_1^2+\epsilon_2^2+\epsilon_3^2}{3\epsilon_1\epsilon_2\epsilon_3\pi}\mu = -\frac{\tau_0^3}{3\pi^4\sigma_3} - \frac{\sigma_2\tau_0}{3\pi^2\sigma_3},
\tag{113}
$$

$$
\langle d_2 d_4 \rangle_c^{\mathrm{GC}} = -\frac{1920}{105\epsilon_1\epsilon_2\epsilon_3\pi}\mu^7 - \frac{1876(\epsilon_1^2+\epsilon_2^2+\epsilon_3^2)}{105\epsilon_1\epsilon_2\epsilon_3\pi}\mu^5 = -\frac{\tau_0^7}{7\pi^8\sigma_3} - \frac{67\sigma_2\tau_0^5}{60\pi^6\sigma_3},
\tag{114}
$$

by replacing $\mu$ with $\frac{\tau_0}{2\pi}$ where $\sigma_2$ and $\sigma_3$ are defined by (97) and (98). Then we get

$$
\left. \frac{\partial^3}{\partial \tau_0^3} \langle d_1 d_1 \rangle_c^{\mathrm{GC}} \right|_{\tau_0=0} = -\frac{2}{\pi^4\sigma_3} = \langle d_1 d_1 d_0 d_0 d_0 \rangle_c^{\mathrm{pert}},
\tag{115}
$$

$$
\left. \frac{\partial}{\partial \tau_0} \langle d_1 d_1 \rangle_c^{\mathrm{GC}} \right|_{\tau_0=0} = -\frac{\sigma_2}{3\pi^2\sigma_3} = \langle d_1 d_1 d_0 \rangle_c^{\mathrm{pert}},
\tag{116}
$$

$$
\left. \frac{\partial^7}{\partial \tau_0^7} \langle d_2 d_4 \rangle_c^{\mathrm{GC}} \right|_{\tau_0=0} = -\frac{720}{\pi^5\sigma_3} = \langle d_2 d_4 d_0 d_0 d_0 d_0 d_0 d_0 d_0 \rangle_c^{\mathrm{pert}},
\tag{117}
$$

$$
\left. \frac{\partial^5}{\partial \tau_0^5} \langle d_2 d_4 \rangle_c^{\mathrm{GC}} \right|_{\tau_0=0} = -\frac{134\sigma_2}{\pi^6\sigma_3} = \langle d_2 d_4 d_0 d_0 d_0 d_0 d_0 \rangle_c^{\mathrm{pert}}.
\tag{118}
$$

In fact, the leading coefficient (115) and the next-to-leading coefficients (116) precisely agree with the numerical results in [43]![8]

In order to verify our results further, we note that the grand canonical connected higher-point functions of the operator $d_1$ can be derived from the grand canonical potential (46) with non-zero FI parameter $\zeta \neq 0$ since it can be viewed as a generating function of the correlation functions of the operator $d_1$. By shifting the FI parameter $\zeta$ by $\frac{\tau_1}{2\pi}$ and expanding the grand canonical potential (46) in powers of $\tau_1$, we can extract the grand canonical connected higher-point functions of the operator $d_1$

$$
J(\mu)_{\zeta \to \zeta + \frac{\tau_1}{2\pi}} = J_0(\mu) + \sum_{\ell=1}^{\infty} \frac{\tau_1^\ell}{\ell!} \overbrace{\langle d_1 \cdots d_1 \rangle}^{\ell}{}_c^{\mathrm{GC}}.
\tag{119}
$$

---

[8]We thank Davide Gaiotto for telling us that our results (117) and (118) agree with his numerical results.

From the expansion (119) we find that the leading and next-to-leading terms of $(k+1)$-point functions of the operator $d_1$ are given by

$$
\langle \overbrace{d_1 \cdots d_1}^{k+1} \rangle_c^{\text{GC}}
$$
$$
= \frac{(-1)^{k+1} i^{k+1} 2}{3\epsilon_1 \epsilon_2 \epsilon_3 \pi^k} (k+1)! \frac{(l-2i\zeta)^{k+2} + (-1)^{k+1}(l+2i\zeta)^{k+2}}{(l^2 + 4\zeta^2)^{k+2}} \mu^3
$$
$$
+ \frac{(-1)^{k+1} i^{k+1}}{12\epsilon_1 \epsilon_2 \epsilon_3 \pi^k} (k+1)! \frac{(l-2i\zeta)^{k+2} + (-1)^{k+1}(l+2i\zeta)^{k+2}}{(l^2 + 4\zeta^2)^{k+2}} (\epsilon_1^2 + \epsilon_2^2 + \epsilon_3^2)\mu + \mathcal{O}(1) . \quad (120)
$$

The result for $k = 0$ precisely agrees with (153) obtained from the formula (93) of the grand canonical one-point function. Also the result for $k = 1$ coincides with (157) obtained from the formula (112) of the grand canonical connected two-point function.

It is obvious to see that the grand canonical one-point functions (93) with non-zero FI parameter $\zeta \neq 0$ can also be viewed as the generating function of the grand canonical connected correlation functions with an insertion of $d_n$ and an arbitrary number of $d_1$. By replacing $\zeta$ with $\zeta + \frac{\tau_1}{2\pi}$ in (93) and then expanding it in powers of $\tau_1$, we find

$$
\frac{\langle d_n \rangle_{\zeta \to \zeta + \frac{\tau_1}{2\pi}}^{\text{GC}}}{\Xi} = \sum_{\ell=0}^{\infty} \frac{\tau_1^\ell}{\ell!} \langle d_n \overbrace{d_1 \cdots d_1}^{\ell} \rangle_c^{\text{GC}} . \quad (121)
$$

Thus we obtain the grand canonical connected correlation functions of $d_n$ and an arbitrary number of $d_1$:

$$
\langle d_n \overbrace{d_1 \cdots d_1}^{k} \rangle_c^{\text{GC}}
$$
$$
= \frac{(-1)^k i^{n+k} 2^{n+1}}{\epsilon_1 \epsilon_2 \epsilon_3 \pi^k} \frac{(n+k)!}{(n+2)!} \frac{((-1)^n (l-2i\zeta)^{n+k+1} + (-1)^k (l+2i\zeta)^{n+k+1})}{(l^2+4\zeta^2)^{n+k+1}} \mu^{n+2}
$$
$$
+ \frac{(-1)^k i^{n+k} 2^{n-5}}{3\epsilon_1 \epsilon_2 \epsilon_3 \pi^k} \frac{(n+k-2)!}{n!} (\epsilon_1^2 + \epsilon_2^2 + \epsilon_3^2)
$$
$$
\times \left\{ (-1)^k \frac{n(n-1)(l-2i\zeta)^2 + 4(n+k-1)(n+k)}{(l-2i\zeta)^{n+k+1}} \right.
$$
$$
\left. + (-1)^n \frac{n(n-1)(l+2i\zeta)^2 + 4(n+k-1)(n+k)}{(l+2i\zeta)^{n+k+1}} \right\} \mu^n + \mathcal{O}(\mu^{n-1}) . \quad (122)
$$

When $n = 1$, we again obtain the connected grand canonical higher-point functions (120) of the operator $d_1$. When $k = 1$, (122) agrees with the formula (112) of the grand canonical connected two-point function for $n_1 = n$ and $n_2 = 1$.

Furthermore, the grand canonical two-point functions (112) of the operators $d_n$ with $\zeta \neq 0$ can be treated as a generating function of the grand canonical higher-point functions with $d_{n_1}$, $d_{n_2}$ and an arbitrary number of $d_1$. By shifting the FI parameter $\zeta$ by $\frac{\tau_1}{2\pi}$ in (112) and expanding

it in powers of $\tau_1$, we obtain the leading and next-to-leading terms of higher-point functions:

$$\langle d_{n_1} d_{n_2} \overbrace{d_1 \cdots d_1}^{k} \rangle_c^{\text{GC}}$$

$$= (-1)^k i^{N_2+k} 2^{N_2} \frac{(N_2+k)!}{(N_2+1)!} \frac{\left[(-1)^{N_2}(l-2i\zeta)^{N_2+k+1} + (-1)^k(l+2i\zeta)^{N_2+k+1}\right]}{\epsilon_1 \epsilon_2 \epsilon_3 \pi^{k+1}(l^2+4\zeta^2)^{N_2+k+1}} \mu^{N_2+1}$$

$$+ \left\{ (-1)^k i^{N_2+k} 2^{N_2-4} \frac{(N_2+k)!}{(N_2-1)!} \frac{\left[(-1)^{N_2}(l-2i\zeta)^{N_2+k+1} + (-1)^k(l+2i\zeta)^{N_2+k+1}\right]}{3\epsilon_1 \epsilon_2 \epsilon_3 \pi^{k+1}(l^2+4\zeta^2)^{N_2+k+1}} \right.$$

$$+ (-1)^k i^{N_2+k} 2^{N_2-6} \frac{(N_2+k-2)!}{(N_2-1)!} \left[n_1(n_1-1) + n_2(n_2-1)\right]$$

$$\times \left. \frac{\left[(-1)^{N_2}(l-2i\zeta)^{N_2+k-1} + (-1)^k(l+2i\zeta)^{N_2+k-1}\right]}{3\epsilon_1 \epsilon_2 \epsilon_3 \pi^{k+1}(l^2+4\zeta^2)^{N_2+k-1}} \right\} (\epsilon_1^2 + \epsilon_2^2 + \epsilon_3^2) \mu^{N_2-1}$$

$$+ \mathcal{O}(\mu^{N_2-2}). \tag{123}$$

Again, when $n_1 = n_2 = 1$, (123) matches with the result (120) of the higher-point functions of $d_1$.

### 3.5.2 Three-point functions

For the grand canonical three-point functions there are contributions from the one-, two- and three-body operators of the forms

$$\mathcal{O}^{(1)}[d_n^3] = \left(\sum_{i=1}^{N} p_n(\sigma_i)\right)^3, \qquad \mathcal{O}^{(2)}[d_n^3] = \sum_{i \neq j} p_n(\sigma_i)^2 p_n(\sigma_j),$$

$$\mathcal{O}^{(3)}[d_n^3] = \sum_{i \neq j \neq k} p_n(\sigma_i) p_n(\sigma_j) p_n(\sigma_k). \tag{124}$$

From the Wigner transforms

$$(\mathcal{O}^{(1)}[d_n^3])_{\text{W}} = p_n(\sigma)^3, \tag{125}$$

$$(P_A \mathcal{O}^{(2)}[d_n^3])_{\text{W}} = 3 p_n(\sigma_1) p_n(\sigma_2)^2 - 3 p_n(\sigma_1)^3 \delta(\sigma_1-\sigma_2)\delta(p_1-p_2) + \cdots \tag{126}$$

$$(P_A \mathcal{O}^{(3)}[d_n^3])_{\text{W}} = p_n(\sigma_1) p_n(\sigma_2) p_n(\sigma_3)$$
$$- 3 p_n(\sigma_1) p_n(\sigma_2)^2 \delta(\sigma_2-\sigma_3)\delta(p_2-p_3)$$
$$+ 2 p_n(\sigma_1)^3 \delta(\sigma_1-\sigma_2)\delta(\sigma_2-\sigma_3)\delta(p_1-p_2)\delta(p_2-p_3) + \cdots \tag{127}$$

of the antisymmetrized operators for (124) where the ellipsis stands for the terms which do not contribute to the leading term, we get the grand canonical three-point functions

$$\frac{\langle d_n d_n d_n \rangle^{\text{GC}}}{\Xi} = \int d\sigma dp (\mathcal{O}^{(1)}[d_n^3])_{\text{W}} \rho_{\text{W}}^{\text{GC}}(\sigma,p)$$

$$+ \int d^2\sigma d^2 p (P_A \mathcal{O}^{(2)}[d_n^3])_{\text{W}} \rho_{\text{W}}^{\text{GC}}(\sigma_1,p_1) \rho_{\text{W}}^{\text{GC}}(\sigma_2,p_2)$$

$$+ \int d^3\sigma d^3 p (P_A \mathcal{O}^{(3)}[d_n^3])_{\text{W}} \rho_{\text{W}}^{\text{GC}}(\sigma_1,p_1) \rho_{\text{W}}^{\text{GC}}(\sigma_2,p_2) \rho_{\text{W}}^{\text{GC}}(\sigma_3,p_3)$$

$$= \int d\sigma dp (p_n(\sigma))^3 \left[ \rho_{\text{W}}^{\text{GC}} - 3(\rho_{\text{W}}^{\text{GC}})^2 + 2(\rho_{\text{W}}^{\text{GC}})^3 \right]$$

$$+ 3 \frac{\langle d_n d_n \rangle^{\text{GC}}}{\Xi} \frac{\langle d_n \rangle^{\text{GC}}}{\Xi} - 2 \left( \frac{\langle d_n \rangle^{\text{GC}}}{\Xi} \right)^3 + \cdots. \tag{128}$$

The grand canonical connected three-point function of the operator $d_n$ is obtained by subtracting the disconnected parts so that the leading term appears from the one-body integral. We obtain

$$
\begin{aligned}
\langle d_n d_n d_n \rangle_c^{\text{GC}} &= \frac{\langle d_n d_n d_n \rangle^{\text{GC}}}{\Xi} - 3 \frac{\langle d_n d_n \rangle^{\text{GC}}}{\Xi} \frac{\langle d_n \rangle^{\text{GC}}}{\Xi} + 2 \left( \frac{\langle d_n \rangle^{\text{GC}}}{\Xi} \right)^3 \\
&= \int d\sigma dp \, (p_n(\sigma))^3 \left[ \rho_{\text{W}}^{\text{GC}} - 3(\rho_{\text{W}}^{\text{GC}})^2 + 2(\rho_{\text{W}}^{\text{GC}})^3 \right] + \cdots \\
&= i^{3n} \epsilon_1^{3n-3} \left[ \frac{1}{\beta^2} \partial_\mu^2 + \frac{\pi^2}{6\beta^4} \partial_\mu^4 + \cdots \right] \int d\sigma dp \left[ \sigma^{3n} + \frac{n(n-1)}{8} \sigma^{3n-2} + \cdots \right] \rho_{\text{W}}^{\text{GC}} \\
&= \frac{i^{3n} 2^{3n-1} \left[ (-1)^n (l-2i\zeta)^{3n+1} + (l+2i\zeta)^{3n+1} \right]}{\epsilon_1 \epsilon_2 \epsilon_3 \pi^2 (l^2 + 4\zeta^2)^{3n+1}} \mu^{3n} + \mathcal{O}(\mu^{3n-2}).
\end{aligned}
\tag{129}
$$

Here we have used the relation

$$
\left( \frac{1}{\beta} \right)^2 \partial_\mu^2 \rho_{\text{W}}^{\text{GC}} = \rho_{\text{W}}^{\text{GC}} - 3(\rho_{\text{W}}^{\text{GC}})^2 + 2(\rho_{\text{W}}^{\text{GC}})^3,
\tag{130}
$$

the Sommerfeld expansion (91) and the result (85). We see that the resulting leading term (129) is invariant under the triality symmetry (2). For $n = 1$ the result (129) agrees with (120).

The next-to-leading term appearing from the one-body integral (129) is not still triality invariant since it involves further contributions from higher order terms in the Wigner transforms (125)-(127). Assuming the triality invariance, we find a consistent expression for the next-to-leading term

$$
\begin{aligned}
\Bigg\{ & i^{3n} 2^{3n-5} (3n)(3n-1) \frac{\left[ (-1)^n (l-2i\zeta)^{3n+1} + (l+2i\zeta)^{3n+1} \right]}{3\epsilon_1 \epsilon_2 \epsilon_3 \pi^2 (l^2 + 4\zeta^2)^{3n+1}} \\
&+ i^{3n} 2^{3n-5} 3n(n-1) \frac{\left[ (-1)^n (l-2i\zeta)^{3n-1} + (l+2i\zeta)^{3n-1} \right]}{3\epsilon_1 \epsilon_2 \epsilon_3 \pi^2 (l^2 + 4\zeta^2)^{3n-1}} \Bigg\} (\epsilon_1^2 + \epsilon_2^2 + \epsilon_3^2) \mu^{3n-2}.
\end{aligned}
\tag{131}
$$

The same argument yields the connected three-point functions for generic three operators $d_{n_1}$, $d_{n_2}$ and $d_{n_3}$

$$
\begin{aligned}
\langle d_{n_1} d_{n_2} d_{n_3} \rangle_c^{\text{GC}} = {} & i^{N_3} 2^{N_3-1} \frac{\left[ (-1)^{N_3} (l-2i\zeta)^{N_3+1} + (l+2i\zeta)^{N_3+1} \right]}{\epsilon_1 \epsilon_2 \epsilon_3 \pi^2 (l^2 + 4\zeta^2)^{N_3+1}} \mu^{N_3} \\
& + \Bigg\{ i^{N_3} 2^{N_3-5} N_3(N_3-1) \frac{\left[ (-1)^{N_3} (l-2i\zeta)^{N_3+1} + (l+2i\zeta)^{N_3+1} \right]}{3\epsilon_1 \epsilon_2 \epsilon_3 \pi^2 (l^2 + 4\zeta^2)^{N_3+1}} \\
& + i^{N_3} 2^{N_3-7} \left[ n_1(n_1-1) + n_2(n_2-1) + n_3(n_3-1) \right] \\
& \times \frac{\left[ (-1)^{N_3} (l-2i\zeta)^{N_3-1} + (l+2i\zeta)^{N_3-1} \right]}{3\epsilon_1 \epsilon_2 \epsilon_3 \pi^2 (l^2 + 4\zeta^2)^{N_3-1}} \Bigg\} (\epsilon_1^2 + \epsilon_2^2 + \epsilon_3^2) \mu^{N_3-2} \\
& + \mathcal{O}(\mu^{N_3-2}),
\end{aligned}
\tag{132}
$$

where $N_3 = n_1 + n_2 + n_3$. We find the triality invariant leading coefficient by analytically computing the one-body integral as in (129). Again the next-to-leading coefficient has additional contributions from the higher order Wigner transforms which we could not analytically evaluate. Instead of computing them explicitly, we obtain a consistent next-to-leading coefficient by restoring the triality invariance.

As a consistency check of our expression (132) of the connected three-point function, note that when one of the three operators is taken as $d_1$, say for $n_3 = 1$, it precisely coincides with the result obtained from (123) when $k = 1$.

We can also obtain the leading and next-to-leading coefficients of the grand canonical connected correlation functions with $d_{n_1}$, $d_{n_2}$, $d_{n_3}$ and an arbitrary number of $d_1$ by shifting the FI parameter in (132) by $\frac{\tau_1}{2\pi}$ and expanding it in powers of $\tau_1$. We find

$$
\langle d_{n_1} d_{n_2} d_{n_3} \overbrace{d_1 \cdots d_1}^{k} \rangle_c^{\text{GC}} =
$$

$$
= (-1)^k i^{n_1+n_2+n_3+k} 2^{N_3-1} \frac{(N_3+k)!}{(N_3)!} \frac{\left[ (-1)^{N_3}(l-2i\zeta)^{N_3+k+1} + (-1)^k(l+2i\zeta)^{N_3+k+1} \right]}{\epsilon_1\epsilon_2\epsilon_3\pi^{k+2}(l^2+4\zeta^2)^{N_3+k+1}} \mu^{N_3}
$$

$$
+ \left\{ (-1)^k i^{N_3+k} 2^{N_3-5} \frac{(N_3+k)!}{(N_3-2)!} \frac{\left[ (-1)^{N_3}(l-2i\zeta)^{N_3+k+1} + (-1)^k(l+2i\zeta)^{N_3+k+1} \right]}{3\epsilon_1\epsilon_2\epsilon_3\pi^{k+2}(l^2+4\zeta)^{N_3+k+1}} \right.
$$

$$
+ (-1)^k i^{N_3+k} 2^{N_3-7} \frac{(N_3+k-2)!}{(N_3-2)!} \left[ n_1(n_1-1) + n_2(n_2-1) + n_3(n_3-1) \right]
$$

$$
\left. \times \frac{(-1)^{N_3}(l-2i\zeta)^{N_3+k-1} + (-1)^k(l+2i\zeta)^{N_3+k-1}}{3\epsilon_1\epsilon_2\epsilon_3\pi^{k+2}(l^2+4\zeta^2)^{N_3+k-1}} \right\} (\epsilon_1^2 + \epsilon_2^2 + \epsilon_3^2)\mu^{N_3-2}
$$

$$
+ \mathcal{O}(\mu^{N_3-3}). \tag{133}
$$

### 3.5.3 Four-point functions

The grand canonical four-point functions can be computed from the one-, two-, three- and four-body operators:

$$
\mathcal{O}^{(1)}[d_n^4] = \left( \sum_{i=1}^{N} p_n(\sigma_i) \right)^4, \qquad \mathcal{O}^{(2)}[d_n^4] = \sum_{i\neq j} p_n(\sigma_i)^3 p_n(\sigma_j) + p_n(\sigma_i)^2 p_n(\sigma_j)^2,
$$

$$
\mathcal{O}^{(3)}[d_n^4] = \sum_{i\neq j\neq k} p_n(\sigma_i)^2 p_n(\sigma_j) p_n(\sigma_k), \quad \mathcal{O}^{(4)}[d_n^4] = \sum_{i\neq j\neq k\neq l} p_n(\sigma_i) p_n(\sigma_j). \tag{134}
$$

The Wigner transforms of the antisymmetrized operators for (134) take the forms

$$
(\mathcal{O}^{(1)}[d_n^4])_{\text{W}} = p_n(\sigma)^4, \tag{135}
$$

$$
\begin{aligned}
(P_{\text{A}}\mathcal{O}^{(2)}[d_n^4])_{\text{W}} = {} & 3p_n(\sigma_1)^2 p_n(\sigma_2)^2 + 4p_n(\sigma_1)^3 p_n(\sigma_2) \\
& - 7p_n(\sigma)^4 \delta(\sigma_1-\sigma_2)\delta(p_1-p_2) + \cdots
\end{aligned} \tag{136}
$$

$$
\begin{aligned}
(P_{\text{A}}\mathcal{O}^{(3)}[d_n^4])_{\text{W}} = {} & 6p_n(\sigma_1)^2 p_n(\sigma_2) p_n(\sigma_3) - 6p_n(\sigma_1)^2 p_n(\sigma_2)^2 \delta(\sigma_2-\sigma_3)\delta(p_2-p_3) \\
& - 12p_n(\sigma)^3 p_n(\sigma_3)\delta(\sigma_1-\sigma_2)\delta(p_1-p_2) \\
& + 12p_n(\sigma_1)^4 \delta(\sigma_1-\sigma_2)\delta(p_1-p_2)\delta(\sigma_2-\sigma_3)\delta(p_2-p_3) + \cdots
\end{aligned} \tag{137}
$$

$$
\begin{aligned}
(P_{\text{A}}\mathcal{O}^{(4)}[d_n^4])_{\text{W}} = {} & p_n(\sigma_1) p_n(\sigma_2) p_n(\sigma_3) p_n(\sigma_4) \\
& - 6p_n(\sigma_1)^2 p_n(\sigma_3) p_n(\sigma_4)\delta(\sigma_1-\sigma_2)\delta(p_1-p_2) \\
& + 3p_n(\sigma_1)^2 p_n(\sigma_3)^2 \delta(\sigma_1-\sigma_2)\delta(p_1-p_2)\delta(\sigma_3-\sigma_4)\delta(p_3-p_4) \\
& + 8p_n(\sigma_1)^3 p_n(\sigma_4)\delta(\sigma_1-\sigma_2)\delta(p_1-p_2)\delta(\sigma_2-\sigma_3)\delta(p_2-p_3) \\
& - 6p_n(\sigma_1)^4 \delta(\sigma_1-\sigma_2)\delta(p_1-p_2)\delta(\sigma_2-\sigma_3)\delta(p_2-p_3)\delta(\sigma_3-\sigma_4)\delta(p_3-p_4) \\
& + \cdots,
\end{aligned} \tag{138}
$$

where the ellipsis indicates the terms which do not affect the leading term.

Making use of (135)-(138), we can compute the grand canonical four-point function of the operator $d_n$

$$\frac{\langle d_n d_n d_n d_n\rangle^{\text{GC}}}{\Xi} =$$

$$= \int d\sigma dp (\mathcal{O}^{(1)}[d_n^4])_{\text{W}} \rho_{\text{W}}^{\text{GC}}(\sigma, p)$$

$$+ \int d^2\sigma d^2 p (P_{\text{A}} \mathcal{O}^{(2)}[d_n^4])_{\text{W}} \rho_{\text{W}}^{\text{GC}}(\sigma_1, p_1) \rho_{\text{W}}^{\text{GC}}(\sigma_2, p_2)$$

$$+ \int d^3\sigma d^3 p (P_{\text{A}} \mathcal{O}^{(3)}[d_n^4])_{\text{W}} \rho_{\text{W}}^{\text{GC}}(\sigma_1, p_1) \rho_{\text{W}}^{\text{GC}}(\sigma_2, p_2) \rho_{\text{W}}^{\text{GC}}(\sigma_3, p_3)$$

$$+ \int d^4\sigma d^4 p (P_{\text{A}} \mathcal{O}^{(4)}[d_n^4])_{\text{W}} \rho_{\text{W}}^{\text{GC}}(\sigma_1, p_1) \rho_{\text{W}}^{\text{GC}}(\sigma_2, p_2) \rho_{\text{W}}^{\text{GC}}(\sigma_3, p_3) \rho_{\text{W}}^{\text{GC}}(\sigma_4, p_4)$$

$$= \int d\sigma dp (p_n(\sigma))^4 \left[ \rho_{\text{W}}^{\text{GC}} - 7(\rho_{\text{W}}^{\text{GC}})^2 + 12(\rho_{\text{W}}^{\text{GC}})^3 - 6(\rho_{\text{W}}^{\text{GC}})^4 \right]$$

$$+ 4 \frac{\langle d_n d_n d_n\rangle^{\text{GC}}}{\Xi} \frac{\langle d_n\rangle^{\text{GC}}}{\Xi} + 3 \left( \frac{\langle d_n d_n\rangle^{\text{GC}}}{\Xi} \right)^2 - 12 \frac{\langle d_n d_n\rangle^{\text{GC}}}{\Xi} \left( \frac{\langle d_n\rangle}{\Xi} \right)^2 + 6 \left( \frac{\langle d_n\rangle^{\text{GC}}}{\Xi} \right)^4 . \quad (139)$$

In particular, when the FI parameter is turned off, the one and three-point functions vanish so that the grand canonical four-point function (139) is simplified as

$$\frac{\langle d_n d_n d_n d_n\rangle^{\text{GC}}_{\zeta=0}}{\Xi} =$$

$$= \int d\sigma dp (p_n(\sigma))^4 \left[ \rho_{\text{W}}^{\text{GC}} - 7(\rho_{\text{W}}^{\text{GC}})^2 + 12(\rho_{\text{W}}^{\text{GC}})^3 - 6(\rho_{\text{W}}^{\text{GC}})^4 \right] + 3 \left( \frac{\langle d_n d_n\rangle^{\text{GC}}}{\Xi} \right)^2 . \quad (140)$$

The leading term that is proportional to $\mu^{4n+2}$ appears from the disconnected term $3(\langle d_n d_n\rangle^{\text{GC}}/\Xi)^2$.

By eliminating the disconnected terms from (139) and using the relation

$$\left( \frac{1}{\beta} \right)^3 \partial_\mu^3 \rho_{\text{W}}^{\text{GC}} = \rho_{\text{W}}^{\text{GC}} - 7(\rho_{\text{W}}^{\text{GC}})^2 + 12(\rho_{\text{W}}^{\text{GC}})^3 - 6(\rho_{\text{W}}^{\text{GC}})^4 , \quad (141)$$

we obtain the grand canonical connected four-point function of the operator $d_n$

$$\langle d_n d_n d_n d_n\rangle^{\text{GC}}_c = \frac{\langle d_n d_n d_n d_n\rangle^{\text{GC}}}{\Xi} - 4 \frac{\langle d_n d_n d_n\rangle^{\text{GC}}}{\Xi} \frac{\langle d_n\rangle^{\text{GC}}}{\Xi}$$

$$- 3 \left( \frac{\langle d_n d_n\rangle^{\text{GC}}}{\Xi} \right)^2 + 12 \frac{\langle d_n d_n\rangle^{\text{GC}}}{\Xi} \left( \frac{\langle d_n\rangle}{\Xi} \right)^2 - 6 \left( \frac{\langle d_n\rangle^{\text{GC}}}{\Xi} \right)^4$$

$$= i^{4n} \epsilon_1^{4n-4} \left[ \frac{1}{\beta^3} \partial_\mu^3 + \frac{\pi^2}{6\beta^5} \partial_\mu^5 + \cdots \right] \int d\sigma dp \left[ \sigma^{4n} + \frac{n(n-1)}{6} \sigma^{4n-2} + \cdots \right] \rho_{\text{W}}^{\text{GC}}$$

$$= \frac{2^{4n} n \left[ (l - 2i\zeta)^{4n+1} + (l + 2i\zeta)^{4n+1} \right]}{\epsilon_1 \epsilon_2 \epsilon_3 \pi^3 (l^2 + 4\zeta^2)^{4n+1}} \mu^{4n-1} + \mathcal{O}(\mu^{4n-3}) . \quad (142)$$

The leading term of the connected four-point function is proportional to $\mu^{4n-1}$. We see that the expression (142) is triality invariant! When $n = 1$, it coincides with the leading term in the previous result (120).

The next-to-leading terms in the one-body integral (142) are not still triality invariant as there are additional contributions from the the higher order terms in the Wigner transforms

(135)-(138). Assuming that the Wigner transforms complete the triality invariance so that it is consistent with (133), we find next-to-leading terms

$$
\left\{ i^{4n}2^{4n-6}(4n)(4n-1)(4n-2)\frac{\left[(-1)^{4n}(l-2i\zeta)^{4n+1}+(l+2i\zeta)^{4n+1}\right]}{3\epsilon_1\epsilon_2\epsilon_3\pi^3(l^2+4\zeta^2)^{4n+1}} \right.
$$
$$
\left. + i^{4n}2^{4n-8}4n(n-1)(4n-2)\frac{\left[(-1)^{4n}(l-2i\zeta)^{4n-1}+(l+2i\zeta)^{4n-1}\right]}{3\epsilon_1\epsilon_2\epsilon_3\pi^3(l^2+4\zeta^2)^{4n-1}} \right\}(\epsilon_1^2+\epsilon_2^2+\epsilon_3^2)\mu^{4n-3}.
$$
(143)

More generally, we get the grand canonical connected four-point functions for $d_{n_1}$, $d_{n_2}$, $d_{n_3}$ and $d_{n_4}$

$$
\langle d_{n_1}d_{n_2}d_{n_3}d_{n_4}\rangle_c^{\mathrm{GC}} = i^{N_4}2^{N_4-2}N_4\frac{\left[(-1)^{N_4}(l-2i\zeta)^{N_4+1}+(l+2i\zeta)^{N_4+1}\right]}{\epsilon_1\epsilon_2\epsilon_3\pi^3(l^2+4\zeta^2)^{N_4+1}}\mu^{N_4-1}
$$
$$
+\left\{ i^{N_4}2^{N_4-6}N_4(N_4-1)(N_4-2)\frac{\left[(-1)^{N_4}(l-2i\zeta)^{N_4+1}+(l+2i\zeta)^{N_4+1}\right]}{3\epsilon_1\epsilon_2\epsilon_3\pi^3(l^2+4\zeta^2)^{N_4+1}} \right.
$$
$$
+ i^{N_4}2^{N_4-8}(N_4-2)\left[\sum_{i=1}^{4}n_i(n_i-1)\right]
$$
$$
\left. \times\frac{\left[(-1)^{N_4}(l-2i\zeta)^{N_4-1}+(l+2i\zeta)^{N_4-1}\right]}{3\epsilon_1\epsilon_2\epsilon_3\pi^3(l^2+4\zeta^2)^{N_4-1}} \right\}(\epsilon_1^2+\epsilon_2^2+\epsilon_3^2)\mu^{N_4-3}
$$
$$
+\mathcal{O}(\mu^{N_4-4}),
$$
(144)

where $N_4 = n_1 + n_2 + n_3 + n_4$. For $n_4 = 1$ the expression (144) reproduces the connected four-point function obtained from (133) for $k = 1$ involving a single $d_1$.

Again we can extract from (144) the leading and next-to-leading terms of the connected higher-point functions with additional insertion of $d_1$

$$
\langle d_{n_1}d_{n_1}d_{n_3}d_{n_4}\overbrace{d_1\cdots d_1}^{k}\rangle_c^{\mathrm{GC}} =
$$
$$
= (-1)^k i^{N_4+k}2^{N_4-2}\frac{(N_4+k)!}{(N_4-1)!}\frac{\left[(-1)^{N_4}(l-2i\zeta)^{N_4+k+1}+(-1)^k(l+2i\zeta)^{N_4+k+1}\right]}{\epsilon_1\epsilon_2\epsilon_3\pi^{k+3}(l^2+4\zeta^2)^{N_4+k+1}}\mu^{N_4-1}
$$
$$
+\left\{ (-1)^k i^{N_4+k}2^{N_4-6}\frac{(N_4+k)!}{(N_4-3)!}\frac{\left[(-1)^{N_4}(l-2i\zeta)^{N_4+k+1}+(-1)^k(l+2i\zeta)^{N_4+k+1}\right]}{3\epsilon_1\epsilon_2\epsilon_3\pi^{k+3}(l^2+4\zeta^2)^{N_4+k+1}} \right.
$$
$$
+ (-1)^k i^{N_4+k}2^{N_4-8}\frac{(N_4+k-2)!}{(N_4-3)!}\left[\sum_i n_i(n_i-1)\right]
$$
$$
\left. \times\frac{\left[(-1)^{N_4}(l-2i\zeta)^{N_4+k-1}+(-1)^k(l+2i\zeta)^{N_4+k-1}\right]}{3\epsilon_1\epsilon_2\epsilon_3\pi^{k+3}(l^2+4\zeta^2)^{N_4+k-1}} \right\}(\epsilon_1^2+\epsilon_2^2+\epsilon_3^2)\mu^{N_4-3}
$$
$$
+\mathcal{O}(\mu^{N_4-4}).
$$
(145)

### 3.5.4 The $k$-point functions

We can compute more general higher-point functions by considering the many-body operators of $d_n$. The grand canonical $k$-point function is obtained by summing over the averages of the Wigner transforms of the antisymmetrized $l$-body operators $P_{\mathrm{A}}\mathcal{O}^{(l)}[d_n^k]$ with $l = 1,\cdots,k$

$$
\frac{\overbrace{\langle d_n\cdots d_n\rangle}^{k}{}^{\mathrm{GC}}}{\Xi} = \sum_{\ell=1}^{k}\left[\int\prod_{i=1}^{\ell}d\sigma_i dp_i\left(P_{\mathrm{A}}\mathcal{O}^{(\ell)}[d_n^k]\right)_{\mathrm{W}}\prod_{i=1}^{\ell}\rho_{\mathrm{W}}^{\mathrm{GC}}(\sigma_\ell,p_\ell)\right].
$$
(146)

From the grand canonical $k$-point function (146) and the lower-point functions, the grand canonical connected $k$-point functions can be recursively computed as

$$\overbrace{\langle d_n \cdots d_n \rangle}^{k}{}_c^{\text{GC}} = \frac{\overbrace{\langle d_n \cdots d_n \rangle}^{k}{}^{\text{GC}}}{\Xi} - \sum_{\ell=1}^{k-1} \binom{k-1}{\ell-1} \overbrace{\langle d_n \cdots d_n \rangle}^{\ell}{}_c^{\text{GC}} \frac{\overbrace{\langle d_n \cdots d_n \rangle}^{k-\ell}{}^{\text{GC}}}{\Xi} . \tag{147}$$

In particular, the leading terms of the grand canonical connected $k$-point functions can be easily evaluated by acting on the one-body integral with the differential operator $\frac{1}{\beta^{k-1}}\partial_\mu^{k-1}$. We find the leading term of the grand canonical connected $k$-point function of the operators $d_{n_i}$, $i = 1, \cdots, k$:

$$\langle d_{n_1} d_{n_2} \cdots d_{n_k} \rangle_c^{\text{GC}} = i^{N_k} \epsilon_1^{N_k-k} \left[ \frac{1}{\beta^{k-1}} \partial_\mu^{k-1} + \frac{\pi^2}{6\beta^{k+1}} \partial_\mu^{k+1} + \cdots \right] \int d\sigma dp$$
$$\times \left[ \sigma^{N_k} + \frac{\sum_{i=1}^{} n_i(n_i-1)}{24} \sigma^{N_k-2} + \cdots \right] \rho_{\text{W}}^{\text{GC}} + \cdots$$
$$= i^{N_k} 2^{N_k-k+2} \frac{N_k!}{(N_k-k+3)!} \frac{\left[ (-1)^{N_k}(l-2i\zeta)^{N_k+1} + (l+2i\zeta)^{N_k+1} \right]}{\epsilon_1 \epsilon_2 \epsilon_3 \pi^{k-1}(l^2+4\zeta^2)^{N_k+1}} \mu^{N_k-k+3}$$
$$+ \cdots, \tag{148}$$

where $N_k = \sum_{i=1}^{k} n_i$. The leading term is proportional to $\mu^{N_k-k+3}$ and its coefficient is proportional to $\frac{1}{\epsilon_1 \epsilon_2 \epsilon_3}$ so that it is invariant under the triality symmetry (2).

The next-to-leading terms appearing from (148) are not yet triality invariant. Provided that these altogether form the triality invariant expression, we get the consistent triality invariant next-to-leading terms of the grand canonical connected $k$-point function of the operators $d_{n_i}$, $i = 1, \cdots, k$ proportional to $\mu^{N_k+1}$:

$$\left\{ i^{\sum_{i=1}^{k} n_i} 2^{N_k-k-2} \frac{N_k!}{(N_k-k+1)!} \frac{\left[ (-1)^{N_k}(l-2i\zeta)^{N_k+1} + (l+2i\zeta)^{N_k+1} \right]}{3\epsilon_1 \epsilon_2 \epsilon_3 \pi^{k-1}(l^2+4\zeta^2)^{N_k+1}} \right.$$
$$+ i^{N_k} 2^{N_k-k-4} \frac{(N_k-2)!}{(N_k-k+1)!} \left[ \sum_{i=1}^{k} n_i(n_i-1) \right]$$
$$\left. \times \frac{\left[ (-1)^{N_k}(l-2i\zeta)^{N_k-1} + (l+2i\zeta)^{N_k-1} \right]}{3\epsilon_1 \epsilon_2 \epsilon_3 \pi^{k-1}(l^2+4\zeta^2)^{N_k-1}} \right\} (\epsilon_1^2 + \epsilon_2^2 + \epsilon_3^2) \mu^{N_k-k+1}$$
$$+ \mathcal{O}(\mu^{N_k-k}). \tag{149}$$

In fact, we have checked that the expressions (148) and (149) reproduce the leading and next-to-leading terms in all the previous results.

It has been numerically found in [43] that the perturbative correlation function has a conjectural pattern

$$\langle d_{n_1} \cdots d_{n_k} \rangle_c^{\text{pert}} = \sum_{m \geq 0} c_{\{n_i\};m} \sigma_2^m \sigma_3^{-\frac{2m}{3}+\frac{1}{3}\sum_{i=1}^{k}(n_i-1)}, \tag{150}$$

in such a way that the non-vanishing terms have a power of $\sigma_3$ greater or equal to $-1$.

By setting $\mu = \frac{\tau_0}{2\pi}$ in our results (146) and (149) and taking the derivatives with respective

to $\tau_0$, we can write from them the perturbative correlation function in [43]:

$$\langle d_{n_1} \cdots d_{n_k} \overbrace{d_0 \cdots d_0}^{N_k-k+3} \rangle_c^{\text{pert}} = \frac{i^{N_k} N_k! \left[(-1)^{N_k}+1\right]}{2\pi^{N_k+2} l^{N_k+1}} \sigma_3^{-1}, \tag{151}$$

$$\langle d_{n_1} \cdots d_{n_k} \overbrace{d_0 \cdots d_0}^{N_k-k+1} \rangle_c^{\text{pert}} = \frac{i^{N_k} \left[4N_k(N_k-1) + l^2 \sum_{i=1}^{k} n_i(n_i-1)\right](N_k-2)! \left[(-1)^{N_k+1}\right]}{48\pi^{N_k} l^{N_k-1}} \sigma_2 \sigma_3^{-1}. \tag{152}$$

This is compatible with the conjectural pattern (150).

## 4 Concluding remarks

In this work, we evaluated the large $N$ correlation functions of the Coulomb branch operators for 3d $\mathcal{N}=4$ ADHM theory in the Fermi-gas formulation. We confirmed that the leading perturbative part has the triality symmetry, expected from the dual twisted M-theory. Interestingly, the full analytic form of the next-to-leading order correction can be fixed by the requirement of this symmetry even though the Fermi-gas computation is technically hard at this order. This idea should be useful in higher order computations.

We remark several related directions. The correlators have been studied in bootstrap program for 3d SCFTs which arise as the IR limit of the effective theory of multiple M2-branes [13,58–61]. It would be nice to address the $d_n$ correlators via the bootstrap analysis to be compared with our results.

The twisted holography [35] (see also [34]) would relate our results to the perturbative calculations around a dominant semi-classical saddle point in the holographic dual five-dimensional holomorphic (symplectic)-topological theory on $AdS_2 \times S^3$. The $k$-point correlators that we have computed would correspond to an amplitude of the Feynman diagram in the 5d Chern-Simons theory with $k$-points insertion on the 1d defect, where the 1d topological quantum mechanics lives. Beyond the perturbative calculations, the triality symmetry may be broken due to the instanton corrections [43]. It would be nice to extend our Fermi-gas analysis by calculating the non-perturbative corrections to the correlators which capture the full geometry normal to $AdS_2 \times S^3$ in $AdS_4 \times S^7$. Also our subleading terms could be used to test the holographic dual with higher derivative corrections as recently studied in [33,62].

The sphere correlators of the Coulomb and Higgs branch operators can be algebraically computed as a sum of the products of the twisted traces over the Verma modules [19]. For the ADHM theory, this sum is taken over a set of $l$ Young diagrams with $N_k$ ($k=1,\cdots,l$) boxes obeying $\sum_{k=1}^{l} N_k = N$. It would be interesting to analyze the large $N$ behavior of the correlators in terms of the twisted traces.

In [43] it is conjectured from the numerical and algebraic computations that the generating function of connected correlation functions of the operator $d_n$ satisfies a recursion relation that leads to a quadratic constraint on the perturbative correlation functions. This is reminiscent of the "string equation" [63,64] in topological gravity and it may play a key role in the twisted holography. It is intriguing to give an analytical derivation or proof of this relation by extending our analysis.

The line operators in 3d $\mathcal{N}=4$ ADHM theory would be also realized in the twisted M-theory by introducing extra M2-branes intersecting with the original ones. The space of the local operators living at junctions of line operators is realized as Hom space which generalizes the bulk Coulomb and Higgs branch algebras [65]. It is interesting to figure out the triality symmetry in the presence of line operators by applying the Fermi-gas analysis as studied for the ABJM model in [53].

Partition functions of 4d $\mathcal{N} = 2$ SQFTs on $S^3 \times S^1$, the superconformal index or its specialization known as the Schur index [66] can reduce to partition functions of 3d $\mathcal{N} = 4$ SQFTs on $S^3$ [67,68]. The Schur index can be decorated by the line operators wrapping the $S^1$ so that it can be associated with the sphere correlators of the Coulomb branch operators for 3d $\mathcal{N} = 4$ SQFTs. It would be interesting to extend our Fermi-gas analysis to the Schur index with line operators for the 4d $\mathcal{N} = 2^*$ gauge theory which reduces to the ADHM theory as in [69] to show the triality symmetry explicitly.

## Acknowledgements

We would like to thank Davide Gaiotto, Jihwan Oh and Kazumi Okuyama for useful discussions and comments. The work of Y.H. is supported by JSPS KAKENHI Grant No. JP18K03657. The work of T.O. is supported by STFC Consolidated Grants ST/P000371/1.

# A  Some explicit results for correlation functions

The general results in the main text are quite complicated. In this appendix, we summarize explicit forms of the connected correlation functions of $d_n$ for some lower $n$'s in the large $\mu$ limit.

For the one-point functions, we have the following leading and next-to-leading perturbative terms:

$$\langle d_1 \rangle_c^{\text{GC}} = -\frac{16l\zeta}{3\epsilon_1\epsilon_2\epsilon_3(l^2+4\zeta^2)^2}\mu^3 - \frac{2l\zeta(\epsilon_1^2+\epsilon_2^2+\epsilon_3^2)}{3\epsilon_1\epsilon_2\epsilon_3(l^2+4\zeta^2)^2}\mu \,, \tag{153}$$

$$\langle d_2 \rangle_c^{\text{GC}} = -\frac{4l(l^2-12\zeta^2)}{3\epsilon_1\epsilon_2\epsilon_3(l^2+4\zeta^2)^3}\mu^4 - \frac{\left[l^5+(8\zeta^2+4)l^3+16l\zeta^2(\zeta^2-3)\right](\epsilon_1^2+\epsilon_2^2+\epsilon_3^2)}{12\epsilon_1\epsilon_2\epsilon_3(l^2+4\zeta^2)^3}\mu^2 \tag{154}$$

$$\langle d_3 \rangle_c^{\text{GC}} = -\frac{64l\zeta(l^2-4\zeta^2)}{5\epsilon_1\epsilon_2\epsilon_3(l^2+4\zeta^2)^4}\mu^5 - \frac{i(\epsilon_1^2+\epsilon_2^2+\epsilon_3^2)\left[l^4+8(\zeta^2+1)l^2+16\zeta^2(\zeta^2-2)\right]}{3\epsilon_1\epsilon_2\epsilon_3(l^2+4\zeta^2)^4}\mu^3 \,, \tag{155}$$

$$\langle d_4 \rangle_c^{\text{GC}} = \frac{32l(l^4-40l^2\zeta^2+80\zeta^4)}{15\epsilon_1\epsilon_2\epsilon_3(l^2+4\zeta^2)^5}\mu^6 + \frac{(\epsilon_1^2+\epsilon_2^2+\epsilon_3^2)}{6\epsilon_1\epsilon_2\epsilon_3}\left[\frac{4+(l-2i\zeta)^2}{(l-2i\zeta)^5}+\frac{4+(l+2i\zeta)^2}{(l+2i\zeta)^5}\right]\mu^4 \,, \tag{156}$$

where $\langle d_n \rangle_c^{\text{GC}} := \langle d_n \rangle^{\text{GC}}/\Xi$.

For the two-point functions, we have:

$$\langle d_1 d_1 \rangle_c^{\text{GC}} = -\frac{8(l^3-12l\zeta^2)}{3\epsilon_1\epsilon_2\epsilon_3\pi(l^2+4\zeta^2)^3}\mu^3 - \frac{(l^3-12l\zeta^2)(\epsilon_1^2+\epsilon_2^2+\epsilon_3^2)}{3\epsilon_1\epsilon_2\epsilon_3\pi(l^2+4\zeta^2)^3}\mu \,, \tag{157}$$

$$\langle d_1 d_2 \rangle_c^{\text{GC}} = \frac{32l\zeta(l^2-4\zeta^2)}{\epsilon_1\epsilon_2\epsilon_3\pi(l^2+4\zeta^2)^4}\mu^4 + \frac{26l\zeta(l^2-4\zeta^2)(\epsilon_1^2+\epsilon_2^2+\epsilon_3^2)}{3\epsilon_1\epsilon_2\epsilon_3\pi(l^2+4\zeta^2)^4}\mu^2 \,, \tag{158}$$

$$\langle d_2 d_2 \rangle_c^{\text{GC}} = \frac{32(l^5-40l^3\zeta^2+80l\zeta^4)}{5\epsilon_1\epsilon_2\epsilon_3\pi(l^2+4\zeta^2)^5}\mu^5 + \frac{26(l^5-40l^3\zeta^2+80l\zeta^4)(\epsilon_1^2+\epsilon_2^2+\epsilon_3^2)}{9\epsilon_1\epsilon_2\epsilon_3\pi(l^2+4\zeta^2)^5}\mu^3 \,. \tag{159}$$

For the three-point functions with no $d_1$ insertions, we have

$$\langle d_2 d_2 d_2 \rangle_c^{\text{GC}} = -\frac{32\left[(l-2i\zeta)^7 + (l+2i\zeta)^7\right]}{\epsilon_1 \epsilon_2 \epsilon_3 \pi^2 (l^2 + 4\zeta^2)^7} \mu^6 - (\epsilon_1^2 + \epsilon_2^2 + \epsilon_3^2)\mu^4 \times \tag{160}$$
$$\times \left[\frac{20\left[(l-2i\zeta)^7 + (l+2i\zeta)^7\right]}{\epsilon_1 \epsilon_2 \epsilon_3 \pi^2 (l^2 + 4\zeta^2)^7} + \frac{\left[(l-2i\zeta)^5 + (l+2i\zeta)^5\right]}{\epsilon_1 \epsilon_2 \epsilon_3 \pi^2 (l^2 + 4\zeta^2)^5}\right].$$

$$\langle d_2 d_3 d_3 \rangle_c^{\text{GC}} = \frac{128\left[(l-2i\zeta)^9 + (l+2i\zeta)^9\right]}{\epsilon_1 \epsilon_2 \epsilon_3 \pi^2 (l^2 + 4\zeta^2)^9} \mu^8 + (\epsilon_1^2 + \epsilon_2^2 + \epsilon_3^2)\mu^6 \times \tag{161}$$
$$\times \left[\frac{448\left[(l-2i\zeta)^9 + (l+2i\zeta)^9\right]}{3\epsilon_1 \epsilon_2 \epsilon_3 \pi^2 (l^2 + 4\zeta^2)^9} + \frac{28\left[(l-2i\zeta)^7 + (l+2i\zeta)^7\right]}{3\epsilon_1 \epsilon_2 \epsilon_3 \pi^2 (l^2 + 4\zeta^2)^7}\right].$$

$$\langle d_4 d_4 d_4 \rangle_c^{\text{GC}} = \frac{2048\left[(l-2i\zeta)^{13} + (l+2i\zeta)^{13}\right]}{\epsilon_1 \epsilon_2 \epsilon_3 \pi^2 (l^2 + 4\zeta^2)^{13}} \mu^{12} + (\epsilon_1^2 + \epsilon_2^2 + \epsilon_3^2)\mu^{10} \times \tag{162}$$
$$\times \left[\frac{5632\left[(l-2i\zeta)^{13} + (l+2i\zeta)^{13}\right]}{\epsilon_1 \epsilon_2 \epsilon_3 \pi^2 (l^2 + 4\zeta^2)^{13}} + \frac{384\left[(l-2i\zeta)^{11} + (l+2i\zeta)^{11}\right]}{\epsilon_1 \epsilon_2 \epsilon_3 \pi^2 (l^2 + 4\zeta^2)^{11}}\right].$$

For the four-point functions with no $d_1$ insertions, we have

$$\langle d_2 d_2 d_2 d_2 \rangle_c^{\text{GC}} = \frac{512\left[(l-2i\zeta)^9 + (l+2i\zeta)^9\right]}{\epsilon_1 \epsilon_2 \epsilon_3 \pi^3 (l^2 + 4\zeta^2)^9} \mu^7 + (\epsilon_1^2 + \epsilon_2^2 + \epsilon_3^2)\mu^5 \times \tag{163}$$
$$\times \left[\frac{448\left[(l-2i\zeta)^9 + (l+2i\zeta)^9\right]}{\epsilon_1 \epsilon_2 \epsilon_3 \pi^3 (l^2 + 4\zeta^2)^9} + \frac{16\left[(l-2i\zeta)^7 + (l+2i\zeta)^7\right]}{\epsilon_1 \epsilon_2 \epsilon_3 \pi^3 (l^2 + 4\zeta^2)^7}\right].$$

$$\langle d_3 d_3 d_3 d_3 \rangle_c^{\text{GC}} = \frac{12288\left[(l-2i\zeta)^{13} + (l+2i\zeta)^{13}\right]}{\epsilon_1 \epsilon_2 \epsilon_3 \pi^3 (l^2 + 4\zeta^2)^{13}} \mu^{11} + (\epsilon_1^2 + \epsilon_2^2 + \epsilon_3^2)\mu^9 \times \tag{164}$$
$$\times \left[\frac{28160\left[(l-2i\zeta)^{13} + (l+2i\zeta)^{13}\right]}{\epsilon_1 \epsilon_2 \epsilon_3 \pi^3 (l^2 + 4\zeta^2)^{13}} + \frac{1280\left[(l-2i\zeta)^{11} + (l+2i\zeta)^{11}\right]}{\epsilon_1 \epsilon_2 \epsilon_3 \pi^3 (l^2 + 4\zeta^2)^{11}}\right].$$

# B  Fermi surface

We give some details on the evaluation of the quantum corrected Fermi surface.

## B.1  Area of the Fermi surface

The area of the quantum corrected Fermi surface of the region I can be evaluated from the equation (38) as

$$
\begin{aligned}
\mathrm{Vol_I} &= \int_{p=p_*^-}^{p=p_*^+} dp \int_{\sigma=0}^{\sigma=\sigma^+(\mu,p)} d\sigma \\
&= \frac{1}{\pi(l-2i\zeta)}\left\{\int_{p_*^-}^{0} dp\left[\frac{2\pi\mu}{\epsilon_1}+\pi(1+2im)p\right]\right. \\
&\quad \left.-\int_{p_*^-}^{0} dp\left[\log(2\cosh\pi p)+\pi p\right]-\frac{\hbar^2\pi^2}{24}(l-2i\zeta)^2\int_{p_*^-}^{0} dp\, T''((p))\right\} \\
&\quad +\frac{1}{\pi(l-2i\zeta)}\left\{\int_{0}^{p_*^+} dp\left[\frac{2\pi\mu}{\epsilon_1}-\pi(1-2im)p\right]\right. \\
&\quad \left.-\int_{0}^{p_*^+} dp\left[\log(2\cosh\pi p)-\pi p\right]-\frac{\hbar^2\pi^2}{24}(l-2i\zeta)^2\int_{0}^{p_*^+} dp\, T''((p))\right\} \\
&= -\frac{3}{4}\frac{\mu^2}{\epsilon_2(\epsilon_1+\epsilon_2)(l-2i\zeta)}-\frac{1}{12(l-2i\zeta)}-\frac{\hbar^2\pi^2}{12}(l-2i\zeta),
\end{aligned} \tag{165}
$$

where we have extended the integration region to infinity up to non-perturbative terms in $\mu$. Similarly, we can calculate the area of the quantum corrected Fermi surface of the region III:

$$
\begin{aligned}
\mathrm{Vol_{III}} &= \int_{p=p_*^-}^{p_*^+} dp \int_{\sigma=\sigma^-(\mu,p)}^{\sigma=0} d\sigma \\
&= -\frac{3}{4}\frac{\mu^2}{\epsilon_2(\epsilon_1+\epsilon_2)(l+2i\zeta)}-\frac{1}{12(l+2i\zeta)}-\frac{\hbar^2\pi^2}{12}(l+2i\zeta).
\end{aligned} \tag{166}
$$

From the equation (41) one finds the quantum corrected area of the region II

$$
\begin{aligned}
\mathrm{Vol_{II}} &= \int_{\sigma=\sigma_*^-}^{\sigma=\sigma_*^+} d\sigma \int_{p=p_*^+}^{p=p^+(\mu,\sigma)} dp \\
&= \frac{l\mu^2}{2\epsilon_1(\epsilon_1+\epsilon_2)(l^2+4\zeta^2)}-\frac{\epsilon_1 l}{24(\epsilon_1+\epsilon_2)}+\frac{l\hbar^2\pi^2(\epsilon_1+\epsilon_2)}{3\epsilon_1}.
\end{aligned} \tag{167}
$$

The quantum corrected area of the region IV can be similarly computed by using the equation (41). We get

$$
\mathrm{Vol_{IV}} = \int_{\sigma=\sigma_*^-}^{\sigma=\sigma_*^+} d\sigma \int_{p=p^-(\mu,\sigma)}^{p=p_+^-} dp = -\frac{l\mu^2}{2\epsilon_1\epsilon_2(l^2+4\zeta^2)}+\frac{\epsilon_1 l}{24\epsilon_2}-\frac{l\hbar^2\pi^2\epsilon_2}{3\epsilon_1}. \tag{168}
$$

## B.2 Average over the Fermi surface

The average over the quantum Fermi surface of the region I is

$$
\text{Vol}_{\text{I}}^{\sigma^n} = \int_{p=p_*^-}^{p=p_*^-} dp \int_{\sigma=0}^{\sigma=\sigma^+(\mu,p)} d\sigma \, \sigma^n
$$

$$
= \frac{1}{(n+1)\pi^{n+1}(l-2i\zeta)^{n+1}} \int_{p_*^-}^{p_*^+} dp \left[ \frac{2\pi\mu}{\epsilon_1} - T(p) - \frac{\hbar^2\pi^2}{24}(l-2i\zeta)^2 T''(p) \right]^{n+1}. \quad (169)
$$

Although it seems difficult to compute the integral (169) explicitly, we do not need to do so. From the Wigner transform (20) that contains $\hbar^2$ corrections, we can only get the correct leading term proportional to $\mu^{n+2}$ and the next-to-leading term proportional to $\mu^n$. The leading and next-to-leading terms which appear from the expression (169) are

$$
\text{Vol}_{\text{I}}^{\sigma^n} = \frac{1}{(n+1)\pi^{n+1}(l-2i\zeta)^{n+1}}
$$

$$
\times \left( \int_{p_*^-}^{0} dp \left[ \frac{2\pi\mu}{\epsilon_1} + \pi(1+2im)p \right]^{n+1} + \int_{0}^{p_*^+} dp \left[ \frac{2\pi\mu}{\epsilon_1} - \pi(1-2im)p \right]^{n+1} \right.
$$

$$
- (n+1)\left(\frac{2\pi\mu}{\epsilon_1}\right)^n \int_{-\infty}^{0} dp \left[ \{\log(2\cosh\pi p) + \pi p\} + \frac{\hbar^2\pi^2}{24}(l-2i\zeta)^2 T''(p) \right]
$$

$$
\left. - (n+1)\left(\frac{2\pi\mu}{\epsilon_1}\right)^n \int_{0}^{\infty} dp \left[ \{\log(2\cosh\pi p) - \pi p\} + \frac{\hbar^2\pi^2}{24}(l-2i\zeta)^2 T''(p) \right] \right).
$$

These integrals can be evaluated exactly, and we find the following large $\mu$ behavior:

$$
\text{Vol}_{\text{I}}^{\sigma^n} = -\frac{2^{n+2}-1}{2\epsilon_1^n \epsilon_2(\epsilon_1+\epsilon_2)(n+1)(n+2)(l-2i\zeta)^{n+1}} \mu^{n+2} - \frac{2^{n-4}\left(4+(l-2i\zeta)^2\right)}{3\epsilon_1^n(l-2i\zeta)^{n+1}} \mu^n. \quad (170)
$$

Similarly, the average over the quantum Fermi surface of the region III leads to the leading and next-to-leading terms:

$$
\text{Vol}_{\text{III}}^{\sigma^n} = -\frac{(-1)^n(2^{n+2}-1)}{2\epsilon_1^n \epsilon_2(\epsilon_1+\epsilon_2)(n+1)(n+2)(l+2i\zeta)^{n+1}} \mu^{n+2} - \frac{(-1)^n 2^{n-4}\left(4+(l+2i\zeta)^2\right)}{3\epsilon_1^n(l+2i\zeta)^{n+1}} \mu^n. \quad (171)
$$

On the other hand, the average over the quantum Fermi surface of the region II takes the form

$$
\text{Vol}_{\text{II}}^{\sigma^n} = \int_{\sigma=\sigma_*^-}^{\sigma=\sigma_*^+} d\sigma \int_{p=p_*^+}^{p=p^+(\mu,\sigma)} dp \, \sigma^n \quad (172)
$$

$$
= \frac{1}{\pi(1-2im)} \int_{\sigma_*^-}^{0} d\sigma \, \sigma^n \left[ \frac{2\pi\mu}{\epsilon_1} - U(\sigma) + \frac{\hbar^2\pi^2}{12}(1-2im)^2 U''(\sigma) \right] - \int_{\sigma_*^-}^{0} d\sigma \, p_*^+ \sigma^n
$$

$$
+ \frac{1}{\pi(1-2im)} \int_{0}^{\sigma_*^+} d\sigma \, \sigma^n \left[ \frac{2\pi\mu}{\epsilon_1} - U(\sigma) + \frac{\hbar^2\pi^2}{12}(1-2im)^2 U''(\sigma) \right] - \int_{0}^{\sigma_*^+} d\sigma \, p_*^+ \sigma^n.
$$

The integral can be evaluated by extending the integration region to infinity according to the formulas

$$
\int_{0}^{\infty} dx \, x^n \log(1+e^{-2\pi x}) = \frac{\Gamma(2+n)\zeta(2+n)}{(n+1)(4\pi)^{n+1}}(2^{n+1}-1), \quad (173)
$$

$$
\int_{0}^{\infty} dx \, x^n \frac{\pi^2}{\cosh^2 \pi x} = \frac{\Gamma(1+n)\zeta(n)}{(4\pi)^{n-1}}(2^{n-1}-1). \quad (174)
$$

We find

$$\text{Vol}_{\text{II}}^{\sigma^n} = \frac{1}{2\epsilon_1^{n+1}(\epsilon_1+\epsilon_2)(n+1)(n+2)}\left(\frac{1}{(l-2i\zeta)^{n+1}}+\frac{(-1)^n}{(l+2i\zeta)^{n+1}}\right)\mu^{n+2}$$
$$+\frac{l(1+(-1)^n)\Gamma(n+1)\left[2(2^n-2)(\epsilon_1+\epsilon_2)^2\pi^2\zeta(n)-3(2^{n+1}-1)\epsilon_1^2\zeta(n+2)\right]}{2^{2n+3}\cdot 3\pi^{n+2}\epsilon_1(\epsilon_1+\epsilon_2)}. \quad (175)$$

We can analogously calculate the average over the region IV:

$$\text{Vol}_{\text{IV}}^{\sigma^n} = -\frac{1}{2\epsilon_1^{n+1}\epsilon_2(n+1)(n+2)}\left(\frac{1}{(l-2i\zeta)^{n+1}}+\frac{(-1)^n}{(l+2i\zeta)^{n+1}}\right)\mu^{n+2}$$
$$-\frac{l(1+(-1)^n)\Gamma(n+1)\left[2(2^n-2)\epsilon_2^2\pi^2\zeta(n)-3(2^{n+1}-1)\epsilon_1^2\zeta(n+2)\right]}{2^{2n+3}3\pi^{n+2}\epsilon_1\epsilon_2}. \quad (176)$$

## C  Formulae

The Wigner transform of the Hamiltonian operator is given by

$$\begin{aligned}
H_{\text{W}}(q,p) &= T+U+\frac{1}{12}[T,[T,U]_\star]_\star+\frac{1}{24}[U,[T,U]_\star]_\star \\
&\quad +\frac{1}{360}[[[[T,U]_\star,U]_\star,U]_\star,T]_\star-\frac{1}{480}[[[[U,T]_\star,U]_\star,T]_\star,U]_\star \\
&\quad +\frac{1}{360}[[[[U,T]_\star,T]_\star,T]_\star,U]_\star+\frac{1}{120}[[[[T,U]_\star,T]_\star,U]_\star,T]_\star \\
&\quad +\frac{7}{5760}[[[[T,U]_\star,U]_\star,U]_\star,U]_\star-\frac{1}{720}[[[[U,T]_\star,T]_\star,T]_\star,T]_\star \\
&= T(p)+U(q)-\frac{\hbar^2}{12}(T'(p))^2U''(q)+\frac{\hbar^2}{24}(U'(q))^2T''(p) \\
&\quad +\frac{\hbar^4}{144}T'(p)T'''(p)U^{(4)}(q)-\frac{\hbar^4}{288}U'(q)U'''(q)T^{(4)}(p) \\
&\quad -\frac{\hbar^4}{240}(U'(q))^2U''(q)(T''(p))^2+\frac{\hbar^4}{60}(T'(p))^2T''(p)(U''(q))^2 \\
&\quad -\frac{\hbar^4}{80}(U'(q))^2U''(q)T'(p)T'''(p)+\frac{\hbar^4}{120}(T'(p))^2T''(p)U'(q)U'''(q) \\
&\quad +\frac{7\hbar^4}{5760}(U'(q))^4T^{(4)}(p)-\frac{\hbar^4}{720}(T'(p))^4U^{(4)}(q). \quad (177)
\end{aligned}$$

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
