# Peer review of "Fermi-gas correlators of ADHM theory and triality symmetry"

_SciPost Physics, doi:SciPost Phys. 12, 005 (2022)_

## Round 1 · Referee Report · Anonymous (Referee 1) · 2021-9-26

Report

In the paper the authors analyzed the $S^3$ partition function of the 3d N=4 supersymmetric U(N) gauge theory with $l$ fundamental matter multiplets and one adjoint matter multiplet which can be interpreted as N M2-branes placed on an omega-deformed background, where the omega deformation parameters are identified with the mass parameter of the adjoint matter multiplet. They demonstrated that the leading and sub-leading part of the grand potential in the limit of large chemical potential, which correspond to the leading and sub-leading part of the large N free energy, is invariant under the triality symmetry in the omega deformation parameters as suggested from the M-theory background. The authors also evaluated some correlation functions of the same theory and confirmed that the leading part of these quantities are also invariant under the triality symmetry. They further determined the explicit expressions of the sub-leading part of the correlation functions by requiring the triality symmetry.

The analyses of the correlation function are completely original results of the paper. Also, although the large N expansion of the $S^3$ partition function was already obtained in a previous research in a dual description by an ABJM-like theory, the re-interpretation in the omega-deformed M-theory is new. For these reasons I recommend this paper to be published.

---

## Round 1 · Referee Report · Jihwan Oh (Referee 2) · 2021-9-28

Report

In this nice and rigorous paper, the authors compute 3-sphere correlation functions of Coulomb branch operators in 3d N=4 ADHM-like gauge theory. The key advance made was an analytical proof of numerical results, which were derived in arXiv:2004.13810. The result further sheds light on the structure of underlying twisted M-theory background and reconfirms triality property of the algebra of operators in a novel way using correlation function. The authors used well-established technique Fermi gas technique masterfully and the result that they obtain is novel and powerful. The paper is expected to generate a new direction in a growing literature of twisted M-theory.

Overall, this paper has top quality. Therefore, without further editing, I recommend to publish it.

---

## Round 1 · Referee Report · Anonymous (Referee 3) · 2021-9-29

Report

The authors consider N=4 3d supersymmetric gauge theory known as ADHM theory on a three-sphere spacetime. The theory has U(N) gauge group, one adjoint and a certain number of fundamental hypermultiplets. Using the known so-called Fermi-gas approach they rewrite the partition function of the theory and certain correlation functions in terms of a system of N non-interacting fermions and consider its large N limit (or, equivalently, the limit of large chemical in the grand canonical ensemble). The authors find agreement with some predictions from the holographically dual description in M-theory (in particular agreement with the triality symmetry that exchanges three complex lines in the spacetime of M-theory) and numerical results from a previous work by other authors.

I believe that the results and the techniques in this manuscript will be interesting to other researchers working on localization in supersymmetric gauge theories, matrix models, AdS/CFT correspondence and related topics. The paper is generally well written. I would like to recommend it for publication.

Requested changes

I have the following minor suggestions which I think can improve the readability of the paper, particularly for non-specialists:

1) From the expressions (2.8) it seems that the Wigner transform of the Hamiltonian $H_W$ is generically complex valued (in particular its classical part) . Later $(2\pi\mu/\epsilon_1-H_W)$ appears as the argument of the functions like Heaviside step function and Dirac delta function, which are ordinarily defined for a real argument only. I think it would be better if the authors add a clarification about interpretation of such expressions.

2) In the beginning of Section 3 the authors use subscript $n_*$. I suggest that the authors add a comment on what is its meaning, what is the range of the sum in (3.1), and why only $n_*=0$ appears in (3.2).

3) In the formulas like (3.52) (similarly in (3.61)) the authors may consider indicating dependence of $\langle \mathcal{O}\rangle$ on $N$ inside the sum more explicitly , otherwise the formula looks a little confusing.

4) I find that the manuscript in some places (for example around pages 12, 18-21) is quite heavy on technical details which are rather elementary (like calculation of integrals). I think moving them to Appendix might make the reading of the paper more enjoyable. But I leave it up to the authors.

---

## Round 2 · Referee Report · Anonymous (Referee 3) · 2021-10-12

Report

The authors have made appropriate modifications. I would like to recommend the manuscript for publication.

---

## Round 2 · Author Response

We would like to thank the referees for carefully reading the draft and pointing out several improvements. 

  1. In the Fermi-gas analysis we take the real Wigner transform of the Hamiltonian $H_W$ by specifying the imaginary FI and mass parameters. However, we expect that the expression with arbitrary FI and mass parameters can be reached by analytic continuation. We have added comment after eq.(2.10). 

  2. The monopole operators are labeled by the GNO charge $\pm A_{a}$, $a=1,\cdots, r$ where $A$ is the cocharacter that specifies an embedding of a $U(1)$ monopole singularity into the gauge group $G$ and $r$ is the rank of $G$. For $G=U(N)$ we have $A=(A_1, \cdots, A_r)\in \mathbb{Z}^r$ and we denote the charge by $n_*$. 

The correlation function of the Coulomb branch operators can be algebraically presented in terms of the twisted traces over the Verma modules of the quantized Coulomb branch algebra. Since the non-trivial twisted traces involving the monopole operators or equivalently shift operators appear only when they simply shift the vector multiplet scalar fields, only some insertion of ``non-periodic part'' without the shift, i.e. $R_{n_*=0}$ in the integrand will lead to distinct Coulomb branch correlation functions with change of residues. 

We have added these clarifications.

  3. To clarify, $\langle \mathcal{O} \rangle$ depends on $N$ as in (3.2) and $\rho_s$ depends on N as in (3.59). So they cannot be factored out. We have explicitly written their dependence on $N$ in (3.44) and (3.51)-(3.53).

  4. We have moved technical details of the calculations to Appendix.

---

## Round 2 · List of Changes

1. additional comment after eq.(2.10). 

  2. clarifications of notations after eq.(3.2)
  3. inclusion of dependence on $N$ in eq.(3.44) and (3.51)-(3.53).
  4. additional Appendix for technical details of the calculations.

---

## Editorial Decision

published